# HardcoreLogic: Challenging Large Reasoning Models with Long-tail Logic Puzzle Games

**Jingcong Liang,**\* **Shijun Wan**\* **& Xuehai Wu**\*
Fudan University
{jcliang22,sjwan25,xhwu25}@m.fudan.edu.cn

**Siyuan Wang**[†]
University of Southern California
sw_641@usc.edu

**Yitong Li, Qianglong Chen & Duyu Tang**
Huawei Technologies Ltd.
{liyitong3,chenqianglong,tangduyu}@huawei.com

**Zhongyu Wei**[†]
Fudan University &
Shanghai Innovation Institute
zywei@fudan.edu.cn

## Abstract

Large Reasoning Models (LRMs) have demonstrated impressive performance on complex tasks, including logical puzzle games that require deriving solutions satisfying all constraints. However, whether they can flexibly apply appropriate rules to varying conditions, particularly when faced with non-canonical game variants, remains an open question. Existing corpora focus on popular puzzles like 9x9 Sudoku, risking overfitting to canonical formats and memorization of solution patterns, which can mask deficiencies in understanding novel rules or adapting strategies to new variants. To address this, we introduce **HardcoreLogic**, a challenging benchmark of over 5,000 puzzles across 10 games, designed to test the robustness of LRMs on the "long-tail" of logical games. HardcoreLogic systematically transforms canonical puzzles through three dimensions: **Increased Complexity (IC)**, **Uncommon Elements (UE)**, and **Unsolvable Puzzles (UP)**, reducing reliance on shortcut memorization. Evaluations on a diverse set of LRMs reveal significant performance drops, even for models achieving top scores on existing benchmarks, indicating heavy reliance on memorized stereotypes. While increased complexity is the dominant source of difficulty, models also struggle with subtle rule variations that do not necessarily increase puzzle difficulty. Our systematic error analysis on solvable and unsolvable puzzles further highlights gaps in genuine reasoning. Overall, HardcoreLogic exposes the limitations of current LRMs and establishes a benchmark for advancing high-level logical reasoning.

**Leaderboard:** https://huggingface.co/spaces/JunsWan/HardcoreLogic
**Dataset:** https://huggingface.co/datasets/xhWu-fd/HardcoreLogic
**Code:** https://github.com/ljcleo/hardcore-logic

## 1 Introduction

Recent large reasoning models (LRMs) (Lin et al., 2025b) have demonstrated remarkable performance across tasks requiring complex reasoning. Among them, logical puzzle games have emerged as a particularly prominent benchmark where models need to deduce or search for solutions to achieve specific goals under logical rules and constraints. Such puzzles probe diverse reasoning skills, including logical deduction (Lin et al., 2025a), pattern recognition (Chollet et al., 2025), and rule induction (Li et al., 2025), while featuring well-defined rules and objectives that enable systematic difficulty control and straightforward evaluation. These characteristics make logical puzzle games an ideal testbed for assessing and advancing LRMs.

---

\*Equal contributors.
[†]Corresponding authors.

Despite recent successes on benchmarks such as Enigmata (Chen et al., 2025a) and ZebraLogic (Lin et al., 2025a), whether LRMs are genuinely capable of true logical reasoning, i.e., flexibly apply appropriate rules to relevant conditions to derive correct conclusions, remains an important question. Take Sudoku as an example: while most real-world puzzles follow the canonical 9x9 format with nine 3x3 zones, variants with alternative constraints or irregular subgrids often prove challenging even for humans. Similarly, existing corpora exhibit a severe imbalance between canonical and non-canonical logic puzzles, making models prone to overfitting to canonical puzzles (Inger et al., 2025), leading to difficulties in solving non-canonical variants that fall into the long-tail of the distribution. This limitation manifests in two specific ways: (1) Models recognize only the canonical form of logical puzzles; when given a variant, they either struggle in understanding the new rules or ignore them, leading to faulty reasoning. (2) Models develop fixed solution strategies and reasoning patterns to solve canonical puzzles; even when they successfully understand the variant, they still apply a mismatched solution strategy, eventually producing errors or being confused.

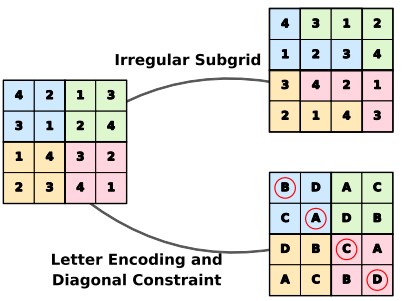 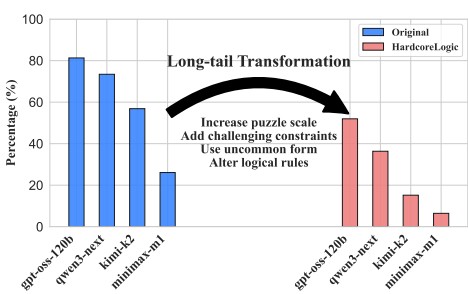

Figure 1: **Left**: Illustrative examples of two long-tail transformed Sudoku. The right top shows an irregular-subgrid Sudoku, replacing standard 2x2 subgrids with irregular subgrids. The bottom right shows a letter-encoded Sudoku with diagonal constraints, where each diagonal must contain all unique symbols. **Right**: Overview of our long-tail transformation applied to logic puzzle games, shows that LRMs have consistent and significant degradation on HardcoreLogic.

Existing logic puzzle benchmarks mainly focus on canonical game forms and fail to expose the aforementioned deficiencies. To address this limitation and provide a detailed inspection of LRMs' reasoning robustness, we introduce **HardcoreLogic**, a logic puzzle game benchmark that challenges models with long-tail variants of puzzles. HardcoreLogic transform common puzzle games along three dimensions: (1) **Increased Complexity (IC)** through larger search spaces and more entangled constraints; (2) **Uncommon Elements (UE)** involving novel rules and altered puzzle forms; (3) **Unsolvable Puzzles (UP)** generated from previously solvable puzzles. The left panel of Figure 1 illustrates two examples of Sudoku transformations designed to increase puzzle difficulty. These enhancements reduce the likelihood that puzzles in HardcoreLogic appear in training corpora, thereby limiting gains from memorizing canonical forms or fixed reasoning patterns.

HardcoreLogic comprises over $5,000$ puzzles spanning 10 logical puzzle games, covering logical deduction, pattern recognition, and sequence searching. Each game is transformed in multiple ways among the three aforementioned dimensions. Comparing with existing datasets of the same games, our puzzles exhibit higher theoretical complexity (for IC puzzles and UE puzzles with novel rules) and higher model perplexity (for UE puzzles with altered forms). Furthermore, our UP puzzles address the absence of unsolvable logical reasoning tasks in mainstream benchmarks.

We evaluate HardcoreLogic across multiple popular and state-of-the-art (SOTA) LRMs, ranging from small distilled models to large open/closed-source models (The right panel of Figure 1 compares the performance of multiple LRMs on the Original and HardcoreLogic). All models, including SOTA models that achieve top performance on baseline benchmarks (e.g., GPT-5), suffer significant performance degradation on HardcoreLogic. Models with stronger reasoning abilities generally exhibit smaller relative drops; however, we also observe large-parameter models that score moderately on the baseline but perform poorly on HardcoreLogic, suggesting the presence of puzzle-game stereotypes in these models. The primary source of difficulty in HardcoreLogic stems from increased complexity, yet we also identified cases where puzzles with novel rules (without added complexity or perplexity) still misled many models. For unsolvable puzzles, models often failed to detect unsolvability and instead produced "partial solutions" that were clearly incorrect.

We further conduct a systematic error analysis to probe the underlying causes of model failures on HardcoreLogic. For solvable puzzles, we classify erroneous responses into six categories, and find that factual errors dominate across models, while more powerful models tend to exhibit brute-force errors, attempting exhaustive searches rather than strategic reasoning. Besides, models' misunderstanding of problem constraints and misapplication of rigid rules lead to significant performance drops. For unsolvable puzzles, our analysis reveals that models performing well on solvable problems genuinely recognize unsolvability better. However, weaker models like Minimax-M1 may output "unsolvable" simply when they fail to find an answer, rather than through true recognition of logical unsatisfiability. When models fail to recognize unsolvability, we observe that stronger models mainly fail due to erroneous reasoning or inability to output answers within token budgets, while weaker models tend to force out solutions even without successfully deriving them. These highlight the need to improve models' deep reasoning capabilities and robustness against degenerate behaviors. Overall, our contributions are threefold:

- We introduce HardcoreLogic, an enhanced benchmark spanning 10 types of logic puzzle games, designed to challenge LRMs with long-tail variants of common puzzle games, featuring higher complexity, novel elements, and unsolvable options.
- We evaluate HardcoreLogic on mainstream and SOTA LRMs, uncovering the limitations of their reasoning abilities. All models, including the latest SOTA models, show substantial performance degradation on puzzles with increased complexity or unfamiliar forms, and exhibit varying behaviors on unsolvable puzzles.
- We conduct a systematic error analysis of LRMs on HardcoreLogic, revealing diverse failure modes and suggesting directions for improving the model's deep reasoning abilities and robustness. In addition, our automatic data construction pipeline provides a scalable protocol for building model training data and environments.

## 2 HARDCORELOGIC

In this section, we introduce the **HardcoreLogic** benchmark, describing the covered logic puzzle types, the long-tail transformation process with statistical analysis, and a detailed complexity analysis of long-tail transformations.

### 2.1 PRELIMINARY: LOGICAL PUZZLE GAMES

In HardcoreLogic, we focus on 10 types of logic games spanning 6 puzzle categories, including 8 challenging subtasks sourced from Enigmata (Chen et al., 2025a), the Zebralogic game from the ZebraLogic dataset (Lin et al., 2025a), and a classic Hanoi game synthesized by ourselves following its standard rules. All these three sources constitute the **Original** data used for comparison with HardcoreLogic. Specifically, HardcoreLogic covers the following 6 categories: (1) **logic puzzle**, (2) **grid puzzle**, (3) **search puzzle**, (4) **pattern puzzle**, (5) **graph puzzle** and (6) **sequential puzzle**. The 10 specific games are **Zebralogic**, **Sudoku**, **Skyscraper**, **Kakurasu**, **Crypto**, **Navigation**, **Binario**, **Minesweeper**, **Hanoi** and **Hitori**. See Appendix B.1 for a more detailed introduction.

### 2.2 LONG-TAIL TRANSFORMATION

Standard logic puzzles are constrained in size, form diversity, and rule design, and thus fail to capture the irregularity and scale of real-world reasoning. To systematically construct more challenging evaluation data, we introduce a set of long-tail transformations that extend puzzles along three distinct dimensions: **Increased Complexity**, **Uncommon Element**, and **Unsolvable Puzzle**.

**Taxonomy** We categorize transformations into five types from three families:

- **Increased Complexity (IC)** enhances difficulty by expanding the search space and depth of reasoning. **Search space expansion (IC1)** enlarges the number of candidate states by reducing the number of initial givens or scaling the puzzle size. For example, removing as many digits as possible while ensuring a unique solution in Binario. **Constraint strengthening (IC2)** increases entanglement among constraints to demand longer reasoning chains. For example, in Zebralogic, instead of *Pet-dog = Sport-football + 1*, we use a looser condition like *Pet-dog > Sport-football*.

- **Uncommon Element (UE)** modifies question forms or rules, often inducing out-of-distribution generalization. **Form variation (UE1)** introduces new types of question forms, such as applying constraints onto irregular subgirds and replacing digits with letters in the Sudoku. **Rule variation (UE2)** alters or hybridizing the governing principles. For example, in Sudoku, we introduce a diagonal constraint requiring that digits on both main diagonals must also be distinct.
- **Unsolvable Puzzle (UP)** deliberately lacks a valid solution, distinguishing them from harder-but-solvable cases. They are used to examine whether large language models can detect inconsistency or insufficiency of information, rather than hallucinate plausible but incorrect answers.

**Basic statistics**  The 10 different logic puzzles in HardcoreLogic have different ways of long-tail transformation types. Table 1 details the aspects of long-tail transformation types that each logic puzzle has. The rules for each logic puzzle and their more specific long-tail transformation details can be found in the Table 3, and each task may correspond to multiple long-tail transformation types.

Table 1: Statistical details of Original and HardcoreLogic on different games and transformations, with the second and last column respectively representing the total sample size of Original and HardcoreLogic on different games. Note that some puzzles belong to multiple transformation categories, so row sums may exceed the overall total.

| Game | Original | Complexity | | Element | | Unsolvable | Overall |
|---|---|---|---|---|---|---|---|
| | | IC1 | IC2 | UE1 | UE2 | | |
| **Zebralogic** | 100 | × | ✓ | × | × | ✓ | 400 |
| **Sudoku** | 100 | ✓ | × | ✓ | ✓ | ✓ | 550 |
| **Skyscraper** | 200 | × | ✓ | × | ✓ | ✓ | 800 |
| **Kakurasu** | 49 | ✓ | ✓ | ✓ | × | ✓ | 300 |
| **Crypto** | 300 | ✓ | × | × | × | ✓ | 400 |
| **Navigation** | 100 | × | ✓ | × | ✓ | ✓ | 300 |
| **Binario** | 150 | ✓ | ✓ | × | × | ✓ | 450 |
| **Hanoi** | 140 | × | × | ✓ | × | ✓ | 800 |
| **Hitori** | 100 | ✓ | × | ✓ | × | ✓ | 500 |
| **Minesweeper** | 150 | ✓ | × | ✓ | ✓ | ✓ | 750 |
| **Overall** | 1389 | 1350 | 1150 | 1400 | 850 | 1350 | 5250 |

## 2.3 Complexity analysis

To systematically evaluate the hardness introduced by our long-tail transformations, we conceptualize complexity as a four-dimensional construct: Search Space Expansion (IC1), Constraint Strengthening (IC2), Form Variation (UE1), and Rule Variation (UE2). Each transformation is associated with dedicated quantitative metrics, and we compare all generated puzzles against the original benchmark. In the following, we present quantitative analyses of these four transformation types to demonstrate how each contributes to increased puzzle difficulty.

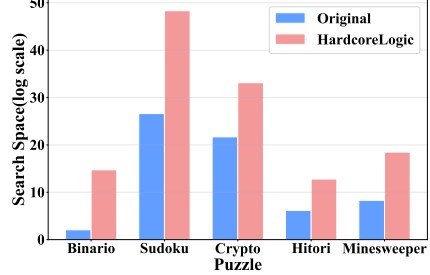

Figure 2: Average search space size (in $\log_{10}$ scale) across five puzzle families. See Appendix B.4 for detailed results.

**Search space expansion (IC1)**  This dimension captures the growth of candidate assignments induced by empty cells. Closed-form formulas are derived for each puzzle family (See Appendix B.4). For instance, in Binario, $N$ empty cells result in a search space of $|S| = 2^N$. Figure 2 shows the average log-scale search space across five puzzle families, confirming that HardcoreLogic systematically enlarges the combinatorial space in all five games.

**Constraint strengthening (IC2)**  This transformation increases puzzle hardness by introducing denser logical entanglement.

- For **CSP-based puzzles** (e.g., Zebralogic, Binario), we encode instances into Z3* and collect: (i) **Decisions**: Explicit branching steps made by the solver; (ii) **Conflicts**: Backtracking events where partial assignments lead to contradictions. Larger counts indicate more complex search spaces and stronger constraint interactions, reflecting higher difficulty.
- For **graph-based puzzles** (Navigation), we apply Dijkstra's algorithm and record: (i) **Generated Nodes**: the number of candidate states created; (ii) **Expanded Nodes**: the number of states fully explored. Increases in both values reflect higher search effort.

As shown in Figure 3 (light green background), HardcoreLogic instances consistently show higher complexity than originals.

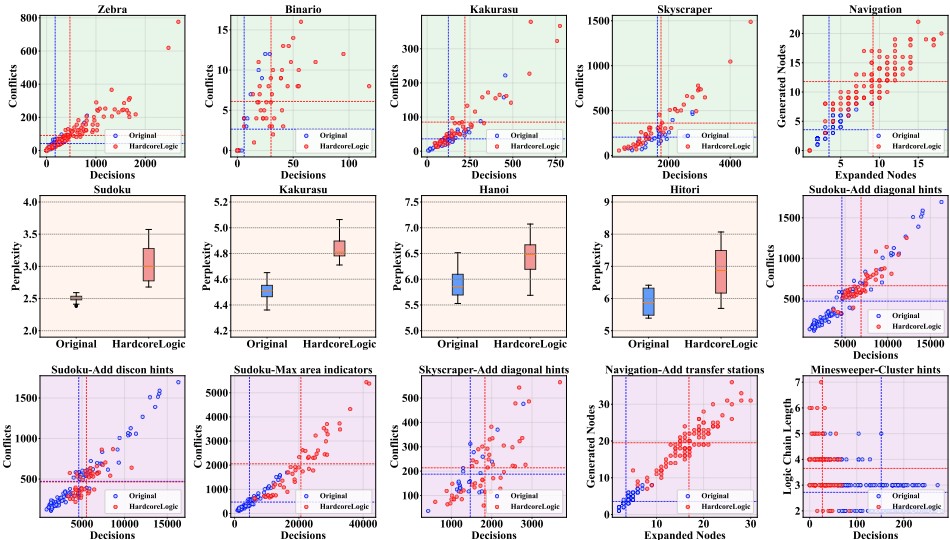

Figure 3: Quantitative comparison of transformation-induced complexity. Panels with light green, light orange, and light purple backgrounds correspond to IC2, UE1, and UE2 , respectively. In each panel, dashed lines indicate the mean value of the corresponding metric.

**Form variation (UE1)**    Form variation introduces novel symbols or forms that preserve puzzle validity but complicate comprehension. Since symbolic solvers cannot capture this representational difficulty, we measure it using **perplexity**, the inverse probability assigned by a pretrained LRM, which quantifies how surprising an instance appears. Higher perplexity values indicate that mutated forms impose greater representational complexity for LRMs. Figure 3 (light orange background) presents boxplots comparing the perplexity distributions of Original and HardcoreLogic instances; form variation consistently results in higher perplexity and thus greater representational difficulty.

**Rule variation (UE2)**    Rule variation modifies or extends the logical rules governing puzzles, forcing solvers to adapt to new structural constraints.

- For **CSP-based puzzles**, we again use Z3 to measure decisions and conflicts.
- For **graph-based puzzles**, we evaluate expanded and generated nodes with Dijkstra's algorithm.

As shown in Figure 3 (light purple background), mutated-rule puzzles consistently yield higher solver statistics, indicating rule changes intensify reasoning complexity. A notable exception is the minesweeper dataset with "landmine clusters": numerical clues now represent adjacent landmine clusters, and more clues are added to ensure unique solutions—this reduces the search space, making required decisions lower than Original. Yet large models show lower accuracy on this modified dataset: unlike counting individual mines, models must continuously track landmine cluster connectivity (e.g., judging cluster membership) for reasoning. This exceeds their simple pattern-matching capabilities, causing performance drops even for powerful models.

---

*Z3 refers to the Satisfiability Modulo Theories (SMT) solver developed by Microsoft Research. (de Moura & Bjørner, 2008)

## 3 EXPERIMENT AND RESULTS

### 3.1 EXPERIMENT SETTINGS

**Benchmark models** We evaluate HardcoreLogic on multiple open-source and closed-source LRMs, a full list available in Appendix C.1. All models except Kimi-K2-Instruct are native LRMs, that is, they support generating a separated reasoning part (usually surrounded by special tokens) before generating the final output. For hybrid reasoning models that can also generate non-reasoning responses (e.g., Qwen3 and DeepSeek-v3.1), we always enable reasoning. For Kimi-K2-Instruct, we guide the model to perform a chain-of-thought (CoT) reasoning. Appendix C.1 also provides details of various model configurations.

**Generation configuration** On open-source models, we limit the reasoning budget to $32,768$ tokens before generating the final answer, regardless of their actual context window limitation. More specifically, we first input the prompt to the model to generate the reasoning part. If the model finishes reasoning within the budget, we then guide the model to generate the final answer that strictly follows the predefined JSON schema to eliminate presentation errors. A generation run is considered correct if and only if the model successfully finishes reasoning and produces a correct answer. We repeat 4 runs on each sample with decoding temperature $T = 0.6$. Closed-source models do not support hard reasoning budget limits, hence we simply limit their total output budget to $32,768$ tokens. Furthermore, we sample 600 cases across all games (5 per transformation type per game) due to expenditure constraints, while remaining repeating 4 runs on each extracted sample. The prompt templates, including corresponding JSON schema for each game, are listed in Appendix C.2.

### 3.2 MAIN RESULTS

**Overall results** Figure 4 illustrates the overall models performance on HardcoreLogic, compared with Original. Kimi-K2-Instruct showed the greatest decrease in accuracy compared to Original on HardcoreLogic. Among open-source models, gpt-oss-120b exhibited the highest accuracy on both datasets, while GPT-5 performed the best in the closed-source models. Minimax-M1 performs the worst among all models.

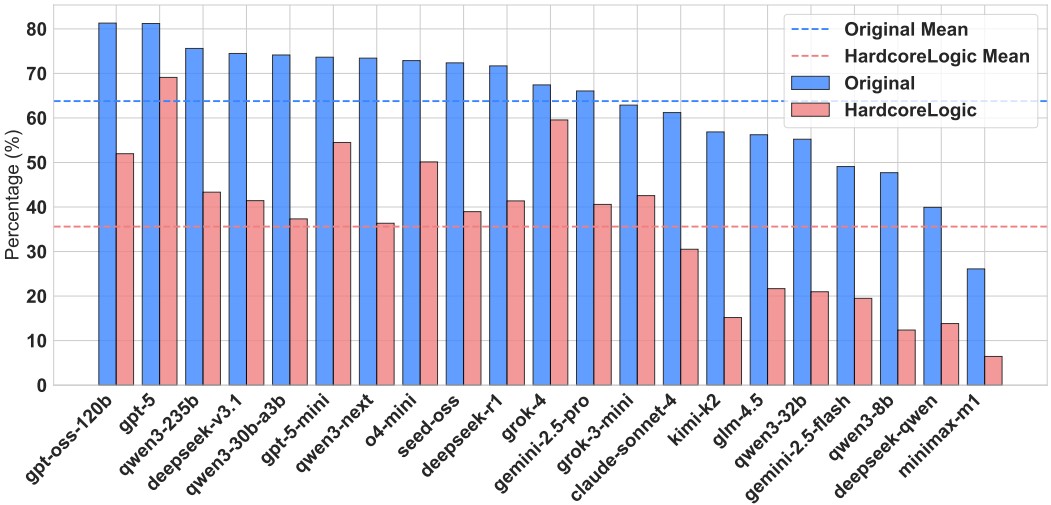

Figure 4: Overall models performance on Original and HardcoreLogic. Dashed lines represent the average values of each model on the corresponding dataset.

**Per-game results** Figure 5 shows the comparison of the accuracy of each puzzle on both Original and the HardcoreLogic across all open-source models.[†] The overall performance of all puzzles and

---
[†]Due to the limited number of subtasks in some puzzles and the small sample size for testing such puzzles on closed-source models, all analysis of per game mainly focuses on open-source models.

models shows a continuous downward trend. For open source models, Binario has the largest average performance degradation on the HardcoreLogic and Original. Skyscraper has the smallest decrease, followed by Navigation. These two puzzles are extremely difficult and extremely simple, which is why they have the smallest decrease.

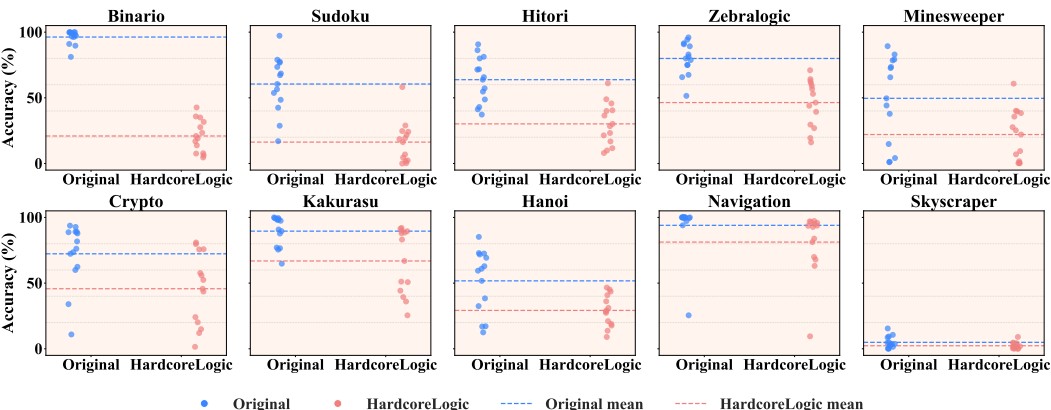

Figure 5: Performance of each puzzle on open-source models.

# 4 ANALYSIS AND DISCUSSION

## 4.1 DIFFERENT LONG-TAIL TRANSFORMATION

In Section 2, we introduce four methods of long-tail transformation, including Search Space Expansion (IC1), Constraint Strengthening (IC2), Form Variation (UE1), and Rule Variation (UE2). Puzzles may also have two different long-tail transformation attributes at the same time. To quantify the impact of different long-tail transformations on puzzle difficulty, we fit a **weighted multiple linear regression** for each puzzle. The dependent variable represents the accuracy of the puzzle after undergoing four different long-tail transformations (IC1, IC2, UE1, UE2). We weight the number of samples in the data, and for mixed transformations, the predictive performance of the model is the sum of its coefficients; the intercept represents the model-predicted baseline accuracy when no long-tail transformation is applied.

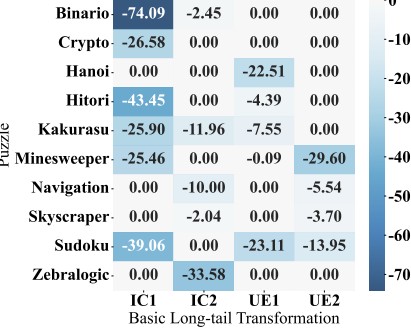

Figure 6: Effects of long-tail transformations on puzzle accuracy.

Figure 6 shows the coefficients of four long-tail transformations in the regression model, which is trained on data from open-source models and reflects their impact on puzzle difficulty. We can observe that **IC1** has the greatest comprehensive impact on the models, as the increase in search space directly requires the improvement of the models' memory and reasoning ability. **UE1** requires the models to recognize some uncommon elements. It is worth noting that the parameter **UE1** reaches its highest value for Sudoku puzzles, mainly due to the need to recognize irregular nine-grid patterns, indicating that the models struggles in this scenario. The parameters of the minesweeper puzzle in **UE2** also show that the "landmine cluster" rule has a significant impact on the models, which is consistent with our hypothesis in Section 2.

## 4.2 ERROR ANALYSIS

To probe the underlying causes of LRM failures on HardcoreLogic, we conduct a systematic error analysis. Based on the comparison between the puzzle, the correct answer, and the model's complete responses, we identify six error categories: (1) **Misunderstanding of the Logic Puzzle**, (2) **Misapplied Solution Framework**, (3) **Brute-Force with Excessive Complexity**, (4) **Factual Errors**, (5) **Over Verfication**, and (6) **Infinite Repetition**. This enables us to move beyond aggregate

accuracy in how different models fail. We randomly sample 50 erroneous cases from each of four representative models: gpt-oss-120b, the best-performing closed-source model on HardcoreLogic; Qwen3-235B, a representative of the Qwen series that we extensively evaluated; Kimi-K2-Instruct, which experienced the largest performance drop from Original to HardcoreLogic; and Minimax-M1, the worst-performing model on HardcoreLogic. We employ GPT-5 (OpenAI, 2025b) as a secondary annotator to classify each case into one of the six categories. Detailed explanation of each category was shown in Appendix C.3.

We employ GPT-5 (OpenAI, 2025b), Gemini-2.5-Pro(Gemini Team, 2025), and Claude-Sonnet-4.5 (Anthropic, 2025) as secondary annotators to classify each case into one of the six categories. The final label is determined through a majority-vote scheme. In situations where the three models produce three distinct labels (i.e., no majority), we conduct manual verification. A detailed consistency analysis of this voting-and-adjudication scheme is provided in the Appendix D.3.

Figure 7 shows the error distribution for each model. Overall,

- **Misunderstanding and Misapplied** errors are particularly prominent in Kimi-K2-Instruct, accounting for roughly $50\%$ of its errors. Notably, Kimi-K2-Instruct also exhibits the largest performance drop from Original to HardcoreLogic, suggesting that this decline is closely associated with its frequent misunderstanding of puzzles and misapplication of solution frameworks. This indicates that the model struggles to correctly interpret problem structures and select appropriate reasoning strategies in more challenging or structurally novel logic problems. Optimization could involve enhancing problem understanding through structured prompts and step-by-step reasoning training, as well as guiding the model to identify problem types and adopt suitable solution frameworks, combined with symbolic or constraint-based verification.
- **Factual** errors are the most prevalent, suggesting that during extended reasoning, LRMs often fabricate facts to fill missing steps, compromising truthfulness and consistency. Mitigation may involve stronger penalties for factual deviations during fine-tuning or reinforcement learning, and mechanisms for intermediate reasoning verification. Meanwhile, during manual review, it was found that the Kimi-K2-Instruct's significant skipping of steps during reasoning makes it more prone to introducing information that is not present in the problem or cannot be directly obtained during the reasoning process, thus making factual type errors more pronounced.
- **Brute-Force** errors in stronger models (gpt-oss-120b, Qwen3-235B) indicate that their generative power can lead to inefficient, enumerative strategies. Performance could be improved by training models to identify problem types and adopt optimal solution frameworks, or by integrating LRM reasoning with symbolic/constraint solvers to guide search.

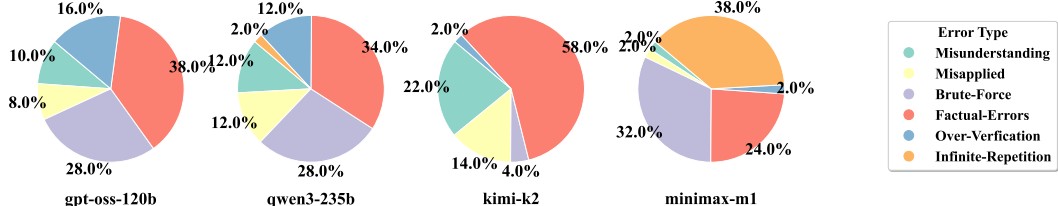

Figure 7: The distribution of error types in HardcoreLogic across the four models.

To compare how model error patterns shift between Original and HardcoreLogic.we also uniformly sample 50 erroneous Original cases and classify them using the identical model-voting and human-verification pipeline. This provides a matched error-type distribution for Baseline errors, enabling a direct comparison against HardcoreLogic results. Figure 8 shows the percentage distribution of six error categories across the two benchmarks.

- **Rule Perturbation Raises Understanding-Related Failures** HardcoreLogic introduces greater rule diversity, non-canonical puzzle structures, and more complex constraint dependencies. These perturbations substantially weaken models' robustness in understanding and applying task rules. As a result, both Misunderstanding and Misapplied errors increase markedly across models.
- **Increased Complexity Reduces Plausible-but-Unfaithful Reasoning** Over-Verification errors decline under HardcoreLogic, indicating that models are less able to generate coherent but incorrect explanations when faced with more complex logical dependencies. Instead of producing confident

and polished but unfaithful rationales, models tend to break earlier in the reasoning process, yielding errors that stem from misunderstanding or rule misapplication.

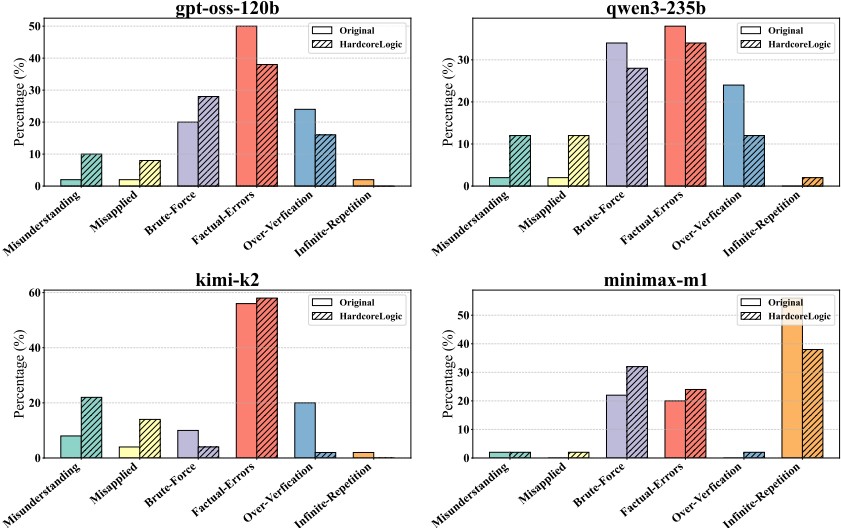

Figure 8: Comparison of error-type distributions between Original and HardcoreLogic across four large language models.

## 4.3 UNSOLVABLE GAMES

**Overall results**   To explore the model's ability to handle contradictory puzzles, we constructed a batch of unsolvable puzzles based on each puzzle. Figure 9 shows the performance of each model in a puzzle-free scenario, where the overall performance of closed-source models is better than that of open-source models. To investigate this phenomenon, we take several open-source models as examples to analyze the possible problems that models may encounter when encountering unsolvable puzzles.

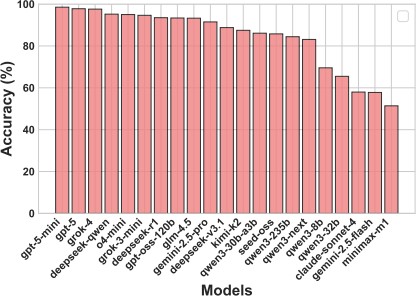

Figure 9: Overall model performance on unsolvable puzzles.

**Sufficiency analysis**   To investigate how LRMs handle unsolvable logic puzzles, we analyzed cases where LRMs correctly labeled puzzles as unsolvable to determine whether the judgment was a genuine understanding (**Justified Unsolvability**) or a heuristic claim due to failure to solve (**Unjustified Unsolvability**). We sampled 50 responses from four models and classified each accordingly, revealing differences in their reasoning behavior, as shown in Figure 10. Stronger models (gpt-oss-120b and Qwen3-235B) typically provide justified explanations, while weaker models (Minimax-M1) more often output unjustified "unsolvable" claims. This suggests that the ability to maintain deeper and more consistent reasoning chains is crucial for producing sufficient unsolvability explanations.

**Error analysis**   We further analyzed the LRMs' incorrect responses to unsolvable logic puzzles, categorizing the errors into four types: (1) **Erroneous Reasoning**, (2) **Mandatory Response**, (3) **Unable to Deduce**, and (4) **Infinite Repetition**. Detailed explanation of each category is given in Appendix C.3. Each incorrect response from the four models was classified accordingly, providing a fine-grained view of how LRMs fail on unsolvable puzzles. As shown in Figure 10, error distributions vary significantly across models. Stronger models (gpt-oss-120b and Qwen3-235B) mainly fail through Erroneous Reasoning or being Unable to Deduce, indicating limitations in sustaining reasoning depth. By contrast, weaker models (Kimi-K2-Instruct and Minimax-M1) exhibit higher rates of Mandatory Responses and especially Infinite Repetition, reflecting brittle control over output structure. These results suggest that future model updates should not only enhance logical

consistency and depth of reasoning but also incorporate stronger mechanisms to prevent degenerate behaviors such as repetitive loops or forced answers.

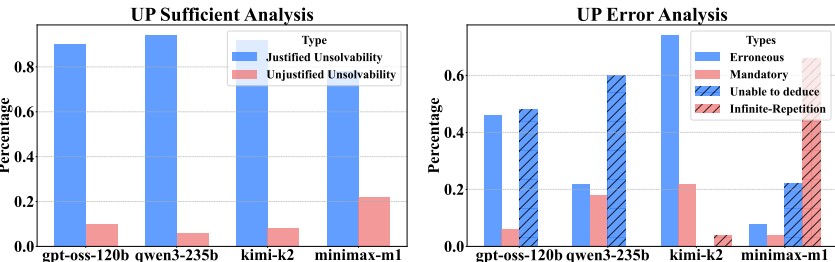

Figure 10: Analysis of LRMs' responses on unsolvable puzzles: the left panel shows correct responses , and the right panel shows incorrect responses.

## 5 RELATED WORK

**Reasoning benchmarks**  Logic puzzles aim to test the logical reasoning ability of a model. Researchers have proposed different benchmarks to test the reasoning ability of models in puzzles, including deductive reasoning (Wang et al., 2022), inductive reasoning, causal reasoning (Yang et al., 2024), and mixed reasoning (Luo et al., 2024). The datasets used include synthetic datasets (Chen et al., 2025a) and collected datasets.Previously, investigators proposed different benchmarks to test and evaluate data sets. For example, Logicgame (Gui et al., 2025) grades the difficulty of tasks by evaluating the number of reasoning steps and achieves dual evaluation of the process and the results. Multi-LogiEval (Patel et al., 2024) systematically evaluated the impact of inference depth on LRM. However, there are still deficiencies in data diversity, with limited difficulty limits for puzzles and a lack of high-difficulty reasoning tasks. We introduced various puzzles, increased the difficulty limit of logical puzzles, performed long-tail transformation on puzzles from multiple aspects, and evaluated the impact of these changes on model performance.

**Long-tail benchmarks**  Several studies have shown that large language models often excel at memorization but struggle to generalize to tasks requiring systematic reasoning or complex combinatorial problem-solving. For example, Anil et al. (2022) and  Wold et al. (2024)highlight that Transformers can fail to generalize to longer sequences or novel compositional structures. These findings suggest that LRMs' apparent reasoning ability may rely heavily on pattern recognition from training data rather than true algorithmic generalization. To systematically evaluate these limitations, several benchmarks have been proposed that target "long-tail" or challenging reasoning instances. For example, JustLogic (Chen et al., 2025b), LINT (Li et al., 2024), and SATbench (Wei et al., 2025) enrich traditional tasks with harder problem instances, extended reasoning chains, or compositional variations, revealing LRMs' difficulty in tackling out-of-distribution or rare configurations. Furthermore, Wang et al. (2025) dynamically generate adversarial questions against LRMs. Building upon these insights, we introduce a new benchmark suite that systematically generates a wide range of logic puzzles under diverse long-tail transformations. Our dataset provides richer structural variations and increased reasoning complexity, allowing a more comprehensive evaluation of LRMs' generalization and problem-solving capacity beyond what prior benchmarks offer.

## 6 CONCLUSION

In this paper, we introduce HardcoreLogic, a challenging logic puzzle benchmark comprising over 5,000 puzzles spanning 10 different puzzle games. Our experiments show that LRMs exhibit a substantial performance drop on HardcoreLogic compared to the Original datasets. This highlights that current models still struggle in less conventional, long-tail scenarios and often rely on pattern recognition or memorized experience rather than genuine reasoning. At the same time, HardcoreLogic provides a valuable benchmark for future research, offering a platform to systematically evaluate and improve the reasoning capabilities of LRMs in diverse and challenging logical contexts.

**Acknowledgments** The work is supported by AI for Science Program, Shanghai Municipal Commission of Economy and Informatization (2025-GZL-RGZN-BTBX-02028). The project's computational resources are partially supported by CFFF platform of Fudan University.

**Ethics statement** Original contains samples from existing published datasets including Enigmata and ZebraLogic, of which we strictly follow the corresponding licenses in data use. Meanwhile HardcoreLogic, we only cover the same logic games but have all puzzles generated independently; we guarantee the transparency and reproducibility of the generation of HardcoreLogic.

**Reproduction statement** We publish both Original and HardcoreLogic to the public for reproduction and future research. We also publish the data generation and evaluation code for reproduction of our datasets and evaluation results. We make our best effort to ensure deterministic outcomes, and guarantee so on open-source LRMs; however, due to the black-box, stochastic nature of closed-source LRMs, we cannot guarantee any precise reproduction on these closed-source models.

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

## A    USE OF LLMS

We use LLMs at several parts of our work: (1) To construct natural-language versions of Zebralogic, we used GPT-4o-mini (OpenAI, 2025c) for clue translation from formal mathematical expressions, with GPT-o4-mini (OpenAI, 2025c) verifying semantic consistency between the mathematical and translated forms. (2) For response annotation, GPT-5 (OpenAI, 2025b), Gemini-2.5-Pro(Gemini Team, 2025), and Claude-Sonnet-4.5 (Anthropic, 2025) as secondary annotators to categorize model responses. (3) During paper preparation, we leveraged GPT-5 to refine our writing by compressing redundant descriptions and improving narrative conciseness. (4) LLMs were also employed to assist in literature search, helping us identify and locate relevant prior work.

## B    BENCHMARK DETAILS

### B.1    CATEGORY DEFINITION

In Section 2, we mention that the HardcoreLogic contains puzzles for 6 different categories. Table 2 provides an introduction to their specific definitions.

### B.2    PUZZLE DEFINITION AND TRANSFORMATIONS

To provide a clearer view of how we constructed the HardcoreLogic, we include additional details on each puzzle type. For every puzzle, we present the *original rules* alongside the *applied transformations*. The original rules specify the standard constraints of the puzzle, while the transformations describe the modifications we introduced to increase reasoning difficulty or adapt the puzzles to our evaluation framework. Table 3 summarizes these rules and transformations, offering a comprehensive reference for reproducibility and further analysis. Figure 11 provides examples of some long-tail transformations for each puzzle.

Table 2: Logic game categories in HardcoreLogic.

| Category | Definition |
|---|---|
| Logic puzzle | This type of puzzle provides us with multiple related logical clues, requiring us to integrate each clue. Zebralogic belong to this category. |
| Grid puzzle | This type of puzzle provides a grid of different sizes, where the cells may be blank cells or cells with numbers. Need to fill in the numbers in blank cells through puzzle rules. Sudoku, skyscraper, and Binario belong to this category. |
| Search puzzle | This type of puzzle requires searching for the required cells through puzzle rules. Minesweeper, Hitori, and Kakurasu belong to this category. |
| Pattern puzzle | This type of puzzle will provide a specific pattern and require us to understand, extract, and apply that pattern. Crypto belong to this category. |
| Graph puzzle | This type of puzzle provides some graphic clues that we need to understand and model to answer questions. Navigation belongs to this category. |
| Sequential puzzle | This type of puzzle requires us to solve a multi-step puzzle in a specific order. Hanoi belongs to this category. |

Table 3: Rules and transformations of each game in HardcoreLogic.

| Puzzle | Rule | Description |
|---|---|---|
| **Sudoku** | Original | **(1) Puzzle categories:** Grid puzzle
**(2) Puzzle rules:**
1. The Sudoku board is a $9 \times 9$ grid, divided into 9 smaller $3 \times 3$ subgrids, which includes known cells (numbers 1–9) and unknown cells.
2. Row constraint: Each row must include all numbers from 1 to 9 without duplication.
3. Column constraint: Each column must include all numbers from 1 to 9 without duplication.
4. Subgrid constraint: Each $3 \times 3$ subgrid must include all numbers from 1 to 9 without duplication.
**(3) Puzzle task:** Fill in numbers into unknown cells to satisfy row, column, and subgrid constraints. |

| Puzzle | Rule | Description |
|---|---|---|
| | Transformation | **(1) More empty cells and larger grid:** Increase the number of unknown cells or use a $16 \times 16$ grid (subgrid $4 \times 4$). 
 **(2) Irregular subgrid:** Divide the $9 \times 9$ grid into nine irregular regions; the three constraints remain unchanged. 
 **(3) Additional constraints:** 
 - Diagonal constraint: Each main and secondary diagonal must include all numbers from 1 to 9 without repetition. 
 - Adjacency constraint: The difference between each cell and its orthogonal neighbors (up, down, left, right) cannot be 1. 
 - Maximize box constraint: Divide the $9 \times 9$ grid into nine $3 \times 3$ subgrids indexed 1–9. Each puzzle requires one subgrid to maximize its score (score = sum of cell index $\times$ cell value). 
 When generating puzzles with additional constraints, we ensure that they have multiple solutions under the original constraints and only one solution under these constraints. This requires more empty cells to ensure this situation. When comparing the difficulty of such puzzles with the Original dataset, we compare the difficulty changes between different constraints and the entire Original dataset. However, their search space did not show a significant improvement compared to the data of **IC1** in **HardcoreLogic**.Therefore, we categorize it as **UE2**. 
 **(4) Letter version:** Replace numbers 1–9 with letters A–I, also applied to irregular, diagonal, and adjacency variants. 
 **(5)Unsolvable puzzle:**In a Sudoku puzzle, a blank cell enters a "no valid number can be filled" state, meaning that the union of its row, column, and range already contains all numbers from 1 to 9. |
| **Kakurasu** | Original | **(1) Puzzle categories:** Search puzzle 
 **(2) Puzzle rules:** 
 1. The Kakurasu board is a $6 \times 6$ grid. Numbers at the top (columns) and on the left (rows) are constraints. 
 2. Row sum = sum of column indices (1-based) of black cells in that row. 
 3. Column sum = sum of row indices (1-based) of black cells in that column. 
 **(3) Puzzle task:** Blacken cells in a $6 \times 6$ grid so that row/column sums equal the given constraints. |
| | Transformation | **(1) Add blocked cells:** Some cells cannot be blackened; grid sizes include $6 \times 6$ and $7 \times 7$. 
 **(2) Hide partial clues:** Some row/column constraints are hidden (denoted by -1); grid sizes include $6 \times 6$ and $7 \times 7$. 
 **(3) Unsolvable puzzle:**First, we generate a solvable puzzle. Then, we modify some of the clues to make the puzzle unsolvable—for example, by swapping certain row or column clues and then verifying that the resulting clues indeed render the puzzle unsolvable. |

| Puzzle | Rule | Description |
|---|---|---|
| **Hitori** | Original | **(1) Puzzle categories:** Search puzzle
**(2) Puzzle rules:**
1. The Hitori board is a 4x4(5x5) grid. Each cell in the grid has a number in the range of 1-4(1-5).
2. Connectivity constraint: All cells that have not been blackened are interconnected(4-connected).
3. Blacked cell constraint: All blackened cells cannot be adjacent(4-connected).
4. Unique constraint: The numbers in each row and column cannot be repeated.
**(3) Puzzle task:** Black some cells in the grid to meet the connectivity constraint, blacked cell constraint, and unique constraint mentioned above. |
| | Transformation | **(1) Larger grid:** Upgrade the grid specifications to 6x6 and 7x7.
**(2) Encrypted:** Encrypt numbers into letters using the following encryption method: In the $i$-th row (from 1 to grid size), the cell numbered $k$ now becomes " 'A'+(i+k-2) % grid size".
**(3) Unsolvable puzzle:** Generate a Hitori puzzle that cannot satisfy all of the original puzzle constraints simultaneously. |
| **Skyscraper** | Original | **(1) Puzzle categories:** Grid puzzle
**(2) Puzzle rules:**
1. The puzzle will be provided with an $n \times n$ sized grid , with blank cells inside. We need to use clues (located around the grid) to fill in the numbers.
2. Definition and constraints for filling in numbers: Each row and column can only be filled with numbers 1 to n (the size of the puzzle), the numbers filled in cannot be repeated, and each number in 1 to n needs to be filled in at least once. The number filled in represents the building height at that location.
3. Clues outside the grid: The clues for each puzzle can be one of the following two ways.
-The count hint: The numbers outside the grid indicate how many buildings can be seen from that direction. Tall buildings will block shorter buildings, and the height of the building is represented by the number filled in the blank cell. At this point, the numbers in the clue indicate how many buildings can be seen from that direction towards the other end (the viewing direction is along the row or column).
-The sum hint: The numbers outside the grid indicate the height of the building visible from that direction. Tall buildings will block shorter buildings, and the height of the building is represented by the number filled in the blank cell. At this point, the numbers in the clue represent the sum of the heights of the buildings that can be seen from that direction towards the other end (the viewing direction is along the row or column).
**(3) Puzzle task:** Fill in numbers in the grid to satisfy Clues outside the grid. |

| Puzzle | Rule | Description |
|---|---|---|
| | Transformation | **(1) Add diagonal constraint:** Add Visibility Clues to the top left, top right, bottom left, and bottom right corners of the grid. The numbers filled in the table need to satisfy additional diagonal constraints in addition to the original four directions of Visibility Clues. But there is no constraint on the diagonal that 1-n cannot be repeatedly filled in. (This constraint only occurs when using the count hint) 
 **(2) Hide partial clues:** Hide clues for certain rows and columns (represented by -1). 
 **(3) Unsolvable puzzle:** First, we generate a set of clues that define a solvable puzzle. Then, we intentionally modify some of the clues to make the puzzle unsolvable. For example, consider a contradiction introduced between the maximum-value clue and the non-minimum clue in a column: the top clue is set to the maximum value (n), while the bottom clue is set to a value between 2 and $n-1$. This is inconsistent because a maximum top clue implies the column must be strictly increasing from 1 to n, in which case the bottom clue should necessarily be 1. |
| **Minesweeper** | Original | **(1) Puzzle categories:** Search puzzle 
 **(2) Puzzle rules:** 
 1. The Minesweeper board is a 9x9 grid. Each cell may be a number 0-8 or a hidden cell (represented by ".", which may contain landmines or not). 
 2. The number in a number cell represents the number of landmines around (8-connected with) it. 
 **(3) Puzzle task:** Search for all cells that can be determined to be landmines through numerical cell clues. |
| | Transformation | **(1) More mines and larger grid:** Increase the number of landmines in the answer or use a 12x12 grid. 
 **(2) Cluster hint:** Convert the meaning of a number cell to the number of landmine clusters surrounding (8-connected with) it. Among the 8 neighbors around a certain grid, any landmines that can be reached by connecting them in 8 directions (up, down, left, right, or diagonal) belong to the same landmine cluster. 
 Since the calculation formula for the original search space of Minesweeper is derived from the rules of the standard Minesweeper, we measure the search space of regional Minesweeper Puzzles as $2^N$, where $N$ denotes the number of unknown cells. The search space of regional Minesweeper Puzzles in **HardcoreLogic** is slightly smaller than that of the **Original**. Therefore, we categorize it as **UE2**. 
 **(3) Letter version:** Use letters A-H to represent numbers 1-8 and Z to represent 0. 
 **(4) Unsolvable puzzle:** First, we generate a puzzle. We then randomly select a non-mine tile and modify its displayed value so that it no longer matches the actual number of surrounding mines. After that, we verify whether this altered value indeed makes the puzzle unsolvable. |

| Puzzle | Rule | Description |
|---|---|---|
| **Binario** | Original | **(1) Puzzle categories:** Grid puzzle
**(2) Puzzle rules:**
1. The Binario board is an n-times-n grid. Each cell may be a number cell (0 or 1) or an empty cell (represented by "."). 
2. Non-adjacent constraint: Each row or column should not have more than two adjacent identical numbers (4-connected).
3. Quantity constraint: The number of 0's and 1's in each row and column is the same.
**(3) Puzzle task:** Fill in the empty cells with the numbers 0 and 1 to satisfy the non-adjacent constraint and quantity constraint. |
| | Transformation | **(1) More empty cells and larger grid:** Increase the number of empty cells or use a larger grid.
**(2) Extra constraint:** Each question will add some unique additional constraints based on the original constraints, which will be described in the specific question.
**(3) Unsolvable puzzle:** First, generate a binario puzzle with only one solution. Then, add a numbered square opposite to the solution, thus making it unsolvable. |
| **Navigation** | Original | **(1) Puzzle categories:** Graph puzzle
**(2) Puzzle rules:**
Each question contains some road signs (such as schools, banks), given some letters, each letter corresponds to a road sign, and one road sign may correspond to several letters. The description of the question provides some one-way paths between letters, as well as a starting letter and an ending landmark.
**(3) Puzzle task:** Find the shortest path from the starting letter to the endpoint landmark. |
| | Transformation | **(1) More complex paths:** Add more complex paths to make the path from the starting point to the endpoint longer.
**(2) Add multi-hop:** Add an intermediate target landmark.
**(3) Unsolvable puzzle:** Destroy the connectivity of the graph so that the starting point cannot reach the ending point. |
| **Hanoi** | Original | **(1) Puzzle categories:** Sequential puzzle
**(2) Puzzle rules:**
1. Hanoi contains m pegs and n disks. Disks are represented by numbers (indicating the size of the disk), pegs are represented by letters, and disks stand above the pegs. Each puzzle will provide the initial state of the peg and disk, as well as the target state.
2. Each step moves a disk on top of a peg to another peg that is either empty or whose current top disk is larger than the moved disk. Pegs cannot be moved.
**(3) Puzzle task:** Move the disks on the Pegs from their initial state to the target state. |

| Puzzle | Rule | Description |
|---|---|---|
| | Transformation | **(1) Random start:** Initially, the disks are randomly distributed on the cylinder under the condition of solvability. **(2) Custom target pegs:** The target pillar is not necessarily the last one, but is randomly assigned. **(3) Custom disk order:** The order of the disks is not necessarily from small to large, but a specified order **(4) Unsolvable puzzle:** The disks are restricted to moving only to the right, and the problem is verified using the BFS algorithm to filter out unsolvable problems. |
| **Crypto** | Original | **(1) Puzzle categories:** Pattern puzzle **(2) Puzzle rules:** 1. For the KPA puzzle: Given a set of plaintext and ciphertext pairs, observe the encryption method to decrypt another ciphertext. 2. For the KKA puzzle: Given an encryption method, decrypt the ciphertext **(3) Puzzle task:** Decrypt the ciphertext to get the plaintext |
| | Transformation | **(1) More random text:** Randomly generate more difficult ciphertext **(2) Two-layer with two samples:** For the KPA puzzle, given two sets of plaintext and ciphertext pairs with the same encryption rules, decrypt the double-encrypted ciphertext **(3) Multiple layers or Multiple segments:** Multi-layer ciphertext encryption or the text will be divided into several parts, and each part will be encrypted using a different encryption method. **(4) Unsolvable Puzzle:** Regarding the KPA decryption issue, two samples are provided, each encrypted using a different method, leading to an unsolvable problem. |
| **Zebralogic** | Original | **(1) Puzzle categories:** Logic Puzzle **(2) Puzzle rules:** 1. Problem scenario: Each problem will describe a scenario that includes a specific number of houses. 2. Characteristics: Specific quantity characteristics (e.g., name, pet, etc.), each feature has a unique item equal to the number of houses. 3. Clues: Each question will provide some clues, including direct correspondence, positional relationships, and other clues. 4. Constraints: No repetition in the same dimension; each house uniquely matches one item from each dimension; reasoning only based on clues **(3) Puzzle task:** Deduce the complete correspondence between houses and all dimensional characteristics based on clues. |
| | Transformation | **(1) Create harder rules:** For example, instead of *Pet-dog = Sport-football + 1*, we use a looser condition like *Pet-dog > Sport-football*. We also add more clue types like "1 of 3" and "imply". **(2) Unsolvable puzzle:** Constructing contradictory constraints that render the problem unsolvable. |

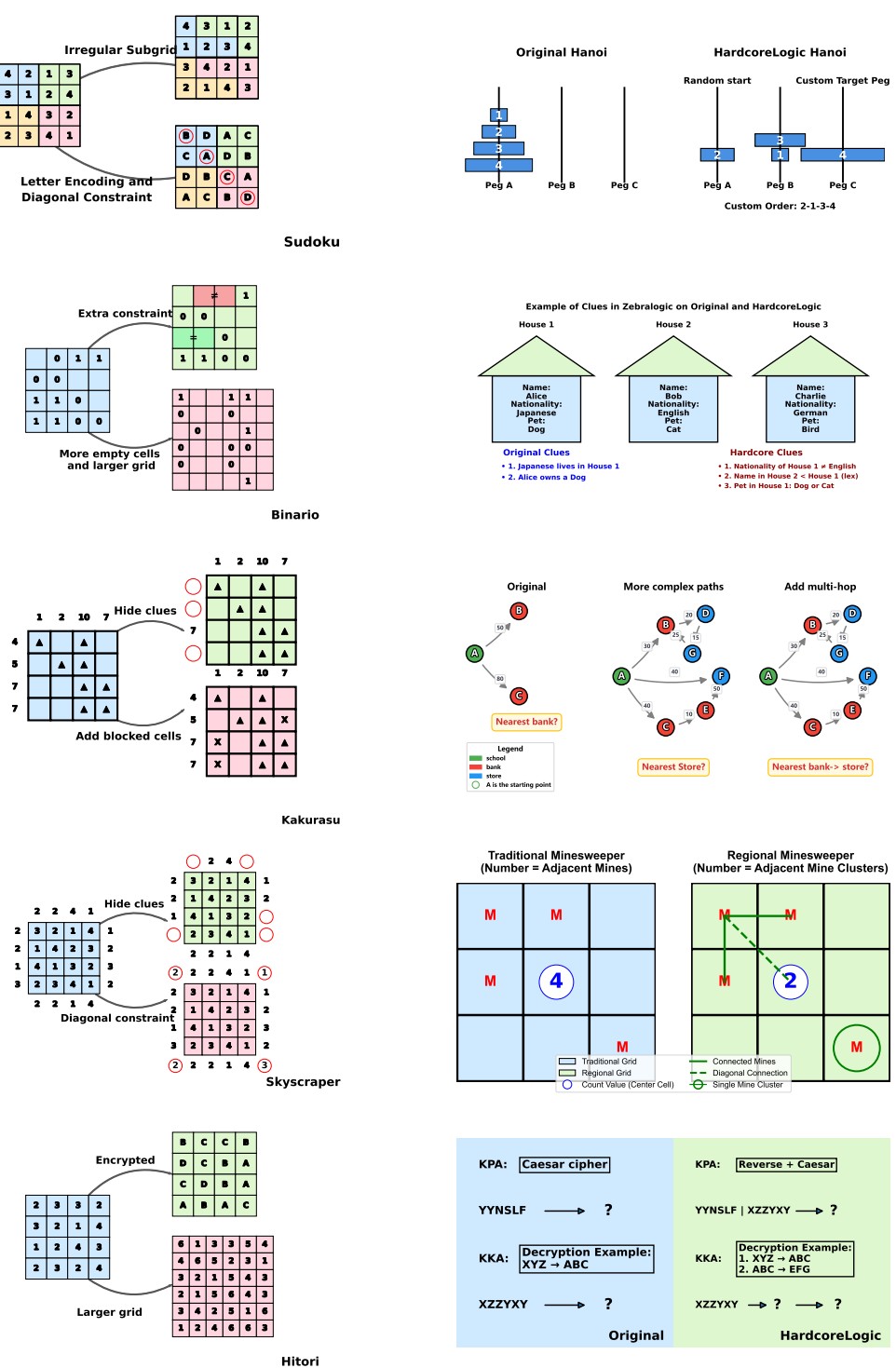

Figure 11: Examples of some long-tail transformations for each puzzle. **Left column**: Examples of Sudoku, Binario, Kakurasu, Skyscraper, and Hitori. **Right column**: Examples of Hanoi, Zebralogic, Navigation, Minesweeper and Crypto.

### B.3 Transformation taxonomy

To systematically characterize the transformations applied in HardcoreLogic, we organize them into a taxonomy spanning three major categories: *increased complexity*, *uncommon element*, and *unsolvable puzzle*. Each category is further divided into subcategories, with representative examples drawn from different puzzle types. This taxonomy (Table 4) illustrates how diverse transformations reshape the puzzles, either by enlarging the search space, strengthening or mutating rules, or deliberately creating contradictions to render puzzles unsolvable.

Table 4: Detailed taxonomy of longtail transformations across puzzles.

| Family | Type | Examples |
|---|---|---|
| **Increased Complexity** | IC1. Search Space Expansion | *Sudoku*: More empty cells and larger grid
*Binario*: More empty cells and larger grid
*Crypto*: More random text
*Crypto*: Multiple layers and segements
*Hitori*: Large grid
*Minesweeper*: More mines and larger grid |
| | IC2. Constraint Strengthening | *Zebralogic*: Create harder rules
*Kakurasu*: Partial hint
*Binario*: Extra constraint
*Skyscraper*: Partial hint
*Navigation*: More complex paths |
| **Uncommon Element** | UE1. Form Variation | *Sudoku*: Letter encoding
*Sudoku*: Irregular subgrid
*Kakurasu*: Add block cells
*Hanoi*: Custom target pegs and disk order
*Hitori*: Encrypted
*Minesweeper*: Letter encoding |
| | UE2. Rule Variation | *Sudoku*: Add diagonal constraint
*Sudoku*: Add Adjacency constraint
*Sudoku*: Maximize box constraint
*Skyscraper*: Add diag hint
*Minesweeper*: Cluster hint
*Navigation*: Add multi-hop |
| **Unsolvable Puzzle** | Unsolvable | *Zebralogic*: Add conflicting constraint
*Sudoku*: Add conflicting hint
*Skyscraper*: Add conflicting constraint
*Kakurasu*: Add conflicting constraint
*Crypto*: Two sample with different encryption method
*Minesweeper*: Add conflicting hint
*Navigation*: Destroy the connectivity of the graph
*Binario*: Add conflicting hints
*Hanoi*: Limit the direction of disk movemen
*Hitori*: Generate an initial solution that does not satisfy the rules |

### B.4 Complexity analysis details

In Section 2.3, we quantify the difficulty of logic puzzles by calculating the search space. Table 5 lists specific expressions for calculating the search space of a logic puzzle, and Table 6 provides detailed results for Figure 2.

Table 5: Definition and formula of search space in logic games.

| Puzzle | Key Parameters | $|S|$ |
|---|---|---|
| Binario | $N$: The number of empty cells | $2^N$ |
| Sudoku | $N$: The number of empty cells 
 $M$: Grid size | $M^N$ |
| Crypto | $L$: Ciphertext length | $26^L$ |
| Minesweeper | $v_i$: Number in digital grid $i$ 
 $N_i$: Number of empty cells around digital grid $i$ | $\prod_i C_{N_i}^{v_i}$ |
| Hitori | $M$: Grid size | $2^{M^2}$ |

Table 6: Search space sizes of puzzles in Table 5 from Original and HardcoreLogic respectively.

| Puzzle | Dataset | Mean | Median |
|---|---|---|---|
| Binario | Original | $1.18 \times 10^2$ | $6.40 \times 10^1$ |
| | HardcoreLogic | $5.43 \times 10^{14}$ | $1.41 \times 10^{14}$ |
| Sudoku | Original | $4.08 \times 10^{26}$ | $6.46 \times 10^{24}$ |
| | HardcoreLogic | $2.10 \times 10^{48}$ | $3.04 \times 10^{52}$ |
| Crypto | Original | $4.97 \times 10^{21}$ | $3.29 \times 10^{20}$ |
| | HardcoreLogic | $1.31 \times 10^{33}$ | $1.35 \times 10^{31}$ |
| Hitori | Original | $1.48 \times 10^6$ | $1.48 \times 10^6$ |
| | HardcoreLogic | $6.22 \times 10^{12}$ | $6.22 \times 10^{12}$ |
| Minesweeper | Original | $2.00 \times 10^8$ | $1.30 \times 10^8$ |
| | HardcoreLogic | $3.03 \times 10^{18}$ | $8.18 \times 10^6$ |

## C EXPERIMENT DETAILS

### C.1 MODEL AND CONFIGURATION

We categorize the LLMs that we use into three types: open-source large models, open-source small models, and closed-source models. Table 7 lists all candidate models with their parameter sizes.

Table 7: Candidate LLMs for experiments. A "closed" size indicates a closed-source model.

| Family | Model | Size |
|---|---|---|
| GPT | GPT-5 (OpenAI, 2025b) | closed |
| | GPT-5 mini (OpenAI, 2025b) | closed |
| | o4 mini (OpenAI, 2025d) | closed |
| Grok | Grok 4 (xAI, 2025b) | closed |
| | Grok 3 mini (xAI, 2025a) | closed |
| Gemini | Gemini 2.5 Pro (Gemini Team, 2025) | closed |
| | Gemini 2.5 Flash (Gemini Team, 2025) | closed |
| Claude | Claude Sonnet 4 (Anthropic, 2025) | closed |
| DeepSeek | DeepSeek-V3.1 (DeepSeek-AI et al., 2024) | 671B |
| | DeepSeek-R1-0528 (DeepSeek-AI et al., 2025) | 671B |
| Qwen | Qwen3-235B-A22B-Thinking-2507 (Yang et al., 2025) | 235B |
| MiniMax | MiniMax-M1-40k (MiniMax et al., 2025) | 456B |
| GLM | GLM-4.5 (GLM-4.5 Team et al., 2025) | 358B |
| Kimi | Kimi-K2-Instruct (Moonshot AI, 2025) | 1T |
| GPT | gpt-oss-120b (OpenAI, 2025a) | 120B |
| DeepSeek | DeepSeek-R1-0528-Qwen3-8B (DeepSeek-AI et al., 2025) | 8B |
| Qwen | Qwen3-Next-80B-A3B-Thinking (Yang et al., 2025) | 80B |
| | Qwen3-32B (Yang et al., 2025) | 32B |
| | Qwen3-30B-A3B-Thinking-2507 (Yang et al., 2025) | 30B |
| | Qwen3-8B (Yang et al., 2025) | 8B |
| Seed | Seed-OSS-36B-Instruct (ByteDance Seed Team, 2025) | 36B |

A few notes:

- We observe in experiments that GPT-5, GPT-5 mini, and o4 mini tend to exceed the $32,768$ token budget more often when choosing the "high" reasoning level. Therefore, we select the "medium" reasoning level to encourage generating valid responses within the limit.
- For gpt-oss-120b, we keep enabling the "high" reasoning level as this model is not prone to the above issue. Following OpenAI's official guidance, we inject this setting by system prompt.
- Kimi-K2-Instruct is not an LRM, hence we ask the model to perform CoT in the system prompt. More specifically, we adopt a two-step generation approach: first, generate a reasoning output wrapped between a pair of special tokens, and then a final answer based on the original prompt and the generated CoT.

### C.2 PROMPT TEMPLATE

To ensure consistency and reproducibility across all puzzle types, we constructed prompt templates using a structured format. Each template specifies the puzzle description, the task instruction, and a standardized JSON output schema. We adopted a Jinja2-style template language so that puzzle instances can be instantiated automatically by substituting parameters such as grid size $n$ and puzzle content. Below, we present the detailed templates for each puzzle family.

**Sudoku Prompt Template**

```
# Puzzle to Solve
{% set n = (subs | length) - 1 %}
```

```
A {{ n }}x{{ n }} sudoku puzzle is a cell grid with {{ n }} rows and {{ n }}
    columns.
The grid is divided into {{ n }} zones, each with {{ n }} cells, outlined with `@
    `.
Each cell contains exactly one of the {{ n }} candidate elements: {% for c in
    subs[1:] %}`{{ c }}`{% if not loop.last %}, {% endif %}{% endfor %}.
The goal is to fill all empty cells (denoted as `.`) with one of these elements.
Each candidate element must appear exactly once in every row.
Each candidate element must appear exactly once in every column.
Each candidate element must appear exactly once in every zone.{% if diag %}
EXTRA: Each candidate element must appear exactly once in the two diagonals.{%
    endif %}{% if discon %}
EXTRA: Adjacent cells cannot have adjacent elements, e.g., `{{ subs[2] }}` and
    `{{ subs[3] }}` cannot be next to each other.{% endif %}{% if irzone %}
WARNING: Zones are NOT regular squares! Pay attention to their outlines!{% elif
    mc_box >= 0 %}
EXTRA: The score of a zone is the sum of `cell_index*cell_value` of all cells in
    the zone,
where cells are indexed as 1 to {{ n }} from left to right, from top to bottom;
the complete puzzle should satisfy that zone {{ mc_box + 1 }} has the highest
    score,
where zones are also indexed from 1 to {{ n }} from left to right, from top to
    bottom.{% endif %}

## Puzzle to Solve
{{ puzzle }}

# Instruction

Now please solve the above sudoku puzzle.
If the puzzle is unsolvable, output `null` as the solution in the following json
    format:

{
"solvable": false,
"solution": null
}

Otherwise, present your solution in the following json format:

{
"solvable": true,
"solution": [
{% for r in range(n) %}[{% for c in range(n) %}"_"{% if c < n - 1 %}, {% endif
    %}{% endfor %}]{% if r < n - 1 %},{% endif %}
{% endfor %}]
}

where each `_` represents the final element in the corresponding cell.
```

---

**Kakurasu Prompt Template**

```
# Puzzle to Solve

A {{ n_row }}x{{ n_col }} kakurasu puzzle is a cell grid with {{ n_row }} rows
    and {{ n_col }} columns.
Rows are numbered 1 to {{ n_row }}, and columns numbered 1 to {{ n_col }}.
The goal is to mark cells to satisfy the following column and row constraints.
On top of the puzzle, a row of {{ n_col }} numbers give the **column**
    constraints --- the row index sum of
```

```
all cells **marked as `O`** in each column; a `-1` indicates that the column has
    no constraint.
At the beginning of each row, a number gives the **row** constraint --- the
    column index sum of
all cells **marked as `O`** in the row; a `-1` indicates that the row has no
    constraint.
The initial grid consists of `.` and `X` cells, and only `.` cells can be marked
    as `O`;
`X` cells **cannot** be marked as `O`.

## Puzzle to Solve
{{ puzzle }}

# Instruction

Now please solve the above kakurasu puzzle.
If the puzzle is unsolvable, output `null` as the solution in the following json
    format:

{
"solvable": false,
"solution": null
}

Otherwise, present your solution in the following json format:

{
"solvable": true,
"solution": [
{% for r in range(n_row) %}[{% for c in range(n_col) %}_{% if c < n_col - 1 %},
    {% endif %}{% endfor %}]{% if r < n_row - 1 %},{% endif %}
{% endfor %}]
}

where each `_` represents whether the corresponding cell is
**marked as `O`** (`true`) or not (`false`).
```

---

**Hitori Prompt Template**

```
# Puzzle to Solve

A {{ n }}x{{ n }} hitori puzzle is a cell grid with {{ n }} rows and {{ n }}
    columns.
The goal is to erase certain cells so that the cells left in each row and in each
    column are unique.
Erased cells cannot be 4-adjacent, and **all** non-erased cells must be 4-
    connected.
A braced cell (`{x}`) cannot be erased, and no more than 3 of its 8-adjacent
    cells can be erased.{% if encrypted %}
WARNING: The puzzle is encrypted into letters!
In row i (from 1 to {{ n }}), a cell with number k now becomes `'A' + (i + k - 2)
    % {{ n }}`.
For example, in row 1 `1` becomes `A`, but in row 2 `1` becomes `B` and `{{ n
    }}` becomes `A`.
Decrypt the puzzle back to numbers before solving it.
{% endif %}

## Puzzle to Solve
{{ puzzle }}

# Instruction
```

```
Now please solve the above hitori puzzle.
If the puzzle is unsolvable, output `null` as the solution in the following json
    format:

{
"solvable": false,
"solution": null
}

Otherwise, present your solution in the following json format:

{
"solvable": true,
"solution": [
{% for r in range(n) %}[{% for c in range(n) %}_{% if c < n - 1 %}, {% endif %}{%
    endfor %}]{% if r < n - 1 %},{% endif %}
{% endfor %}]
}

where each `_` represents whether the corresponding cell is **erased** (`true`)
    or not (`false`).
```

## Skyscraper Prompt Template

```
# Puzzle to Solve

A {{ n }}x{{ n }} skyscraper puzzle is a cell grid with {{ n }} rows and {{ n }}
    columns.
Each cell contains exactly one of the numbers 1 to {{ n }}, representing the "
    height" of the cell.
Each number must appear exactly once in every row and every column.
Looking from a side, a cell in the front blocks **all** cells **behind** it that
    are **not taller**.
The hint of a row/column/diagonal looking from a side is the {{ vv }} of cells
in the row/column/diagonal that are not blocked; a number of `-1` means no
    constraint.
On top of the puzzle, there is a row of {{ n + 2 }} numbers:
the first number is the hint of the main diagonal looking from top left;
the next {{ n }} numbers are the hints of the columns looking from the top;
the last number is the hint of the sub diagonal looking from top right.
Then, at the beginning of each grid row is the hint of that row looking from the
    left;
at the end of that row is the hint of that row looking from the right.
Finally, below the puzzle, there is a row of {{ n + 2 }} numbers:
the first number is the hint of the sub diagonal looking from bottom left;
the next {{ n }} numbers are the hints of the columns looking from the bottom;
the last number is the hint of the main diagonal looking from bottom right.

## Puzzle to Solve
{{ puzzle }}

# Instruction

Now please solve the above skyscraper puzzle.
If the puzzle is unsolvable, output `null` as the solution in the following json
    format:

{
"solvable": false,
"solution": null
```

```
}

Otherwise, present your solution in the following json format:

{
"solvable": true,
"solution": [
{% for r in range(n) %}[{% for c in range(n) %}_{% if c < n - 1 %}, {% endif %}{%
     endfor %}]{% if r < n - 1 %},{% endif %}
{% endfor %}]
}

where each `_` represents the final number in the corresponding cell.
```

## Minesweeper Prompt Template

```
# Puzzle to Solve

A {{ row }}x{{ col }} minesweeper puzzle is a cell grid with {{ row }} rows and
     {{ col }} columns.
Each cell has either one mine (mine cell) or no mine (safe cell).
Some safe cells are opened beforehand, showing the number of
{% if regional %}**8-connected components** of {% endif %}mine cells in their 8-
     adjacent cells.{% if regional %}
For example, if an opened safe cell has three 8-adjacent mine cells,
but all three mine cells are 8-connected with each other,
then the opened safe sell will show `1` instead of `3`.{% endif %}
The goal is to find out all closed cells that must be mine cells.
The puzzle is unsolvable if and only if the current numbers lead to a
     contradiction.{% if no_adj %}
EXTRA: It is also guaranteed that no mines are 8-adjacent to each other.{% endif
     %}{% if letter %}
EXTRA: The puzzle is encrypted into letters, where Z represents 0 and A-H
     represents 1-8.{% endif %}

## Puzzle to Solve
{{ puzzle }}

# Instruction

Now please solve the above minesweeper puzzle.
If the puzzle is unsolvable, output `null` as the solution in the following json
     format:

{
"solvable": false,
"solution": null
}

Otherwise, present your solution in the following json format:

{
"solvable": true,
"solution": [
{% for r in range(row) %}[{% for c in range(col) %}_{% if c < col - 1 %}, {%
     endif %}{% endfor %}]{% if r < row - 1 %},{% endif %}
{% endfor %}]
}

where each `_` represents whether the corresponding cell
**must be a mine cell** (`true`) or safe/undetermined (`false`).
```

**Binario Prompt Template**

```
# Puzzle to Solve

A {{ n }}x{{ n }} binario puzzle is a cell grid with {{ n }} rows and {{ n }}
    columns.
Each cell can either be `0` or `1`.
The goal is to fill all empty cells (denoted as `.`) with `0` or `1`.
Each row must have the same number of `0`s and `1`s.
Each column must have the same number of `0`s and `1`s.
Furthermore, no more than two identical digits are adjacent.

## Puzzle to Solve
{{ puzzle }}

# Instruction

Now please solve the above star battle puzzle.
If the puzzle is unsolvable, output `null` as the solution in the following json
    format:

{
"solvable": false,
"solution": null
}

Otherwise, present your solution in the following json format:

{
"solvable": true,
"solution": [
{% for r in range(n) %}[{% for c in range(n) %}_{% if c < n - 1 %}, {% endif %}{%
    endfor %}]{% if r < n - 1 %},{% endif %}
{% endfor %}]
}

where each `_` represents the final element in the corresponding cell.
```

**Hanoi Prompt Template**

```
# Puzzle to Solve

A {{ n_peg }}x{{ n_disk }} hanoi puzzle has {{ n_peg }} pegs and {{ n_disk }}
    disks.
The disks, in the order of size, are: (smallest) {% for c in order %}`{{ c }}`{%
    if not loop.last %}, {% endif %}{% endfor %} (largest).
The goal is to transform the start state to the goal state in minimum number of
    steps.
Each step moves a disk on top of a peg to another peg that is either empty,
or whose current top disk is larger than the moved disk.{% if right_only %}
Furthermore, the target peg must be to the right of the source peg.{% endif %}

## Puzzle to Solve
{{ puzzle }}

# Instruction

Now please solve the above hanoi puzzle.
If the puzzle is unsolvable, output `null` as the solution in the following json
    format:

{
```

```
"solvable": false,
"solution": null
}

Otherwise, present your solution in the following json format:

{
"solvable": true,
"solution": [
["_", "_"], ["_", "_"], ["_", "_"]...
]
}

where each `["_", "_"]` pair represents the source peg and the target peg of a
      disk-moving step.
```

## Crypto Prompt Template

```
# Puzzle to Solve

An uppercase ASCII text is encrypted into a cipher.
The goal is to recover the plain text, which may or may not have semantic
      meanings.
A list of candidate encryption methods may be provided, one method per line,
in which case the encryption is done by applying each method once sequentially
{% if ordered %} in the given order{% else %}, but NOT necessarily in the given
      order{% endif %}.
Sample plain text-cipher pairs that use the same encryption procedure may also be
       given as a hint.
When "|" appears in the cipher, the encryption is segmented,
where each encryption method consist of multiple sub-methods concatenated with
      "+" in one line,
each applied to the corresponding cipher segment separated by "|".{% if
      prompt_example %}
**IMPORTANT: The encryption method may NOT be the same as in the examples!**
**Use the information below (NOT the examples) to find out the actual encryption
      method!**{% endif %}

## Cipher to Solve

{{ puzzle }}

# Instruction

Now please recover the above cipher.
If the cipher cannot be recovered, e.g. there is a contradiction in the clues,
output `null` as the solution in the following json format:

{
"solvable": false,
"solution": null
}

Otherwise, present your solution in the following json format:

{
"solvable": true,
"solution": "_"
}

where `"_"` represents the plain text string in uppercase.
```

**Zebralogic Prompt Template**

```
# Puzzle to Solve

{{ puzzle }}

# Instruction

Now please solve the above puzzle.
If the puzzle is unsolvable, output `null` as the solution in the following json
    format:

{
"solvable": false,
"solution": null
}

Otherwise, present your solution in the following json format:

{
"solvable": true,
"solution": {
{% for id in house_ids %}"{{ house_alias }} {{ id }}": {
{% for key in keys %}"{{ key }}": "_"{% if not loop.last %},{% endif %}
{% endfor %}}{% if not loop.last %},{% endif %}
{% endfor %}}
}

where each `"_"` represents an attribute in lowercase.
```

**Navigation Prompt Template**

```
# Puzzle to Solve

{{ puzzle }}

# Instruction

Now please solve the above puzzle.
If there is no path, output `null` as the solution in the following json format:

{
"solvable": false,
"solution": null
}

Otherwise, present your solution in the following json format:

{
"solvable": true,
"solution": ["_", ...]
}

where each `"_"` represents a point on the path (an uppercase letter),
including the start point and the end point.
```

## C.3 ERROR TYPES

In Sections 4.2 and 4.3, we specifically classified the types of errors returned by the model. Tables 8 and 9 give specific definitions of each category.

Table 8: Error types in error analysis.

| Category | Definition |
|---|---|
| Misunderstanding | The model does not truly understand the logical puzzle, or there is a deviation in its understanding. |
| Misapplied | The problem was correctly understood, but an inappropriate and often more common thinking framework was applied when selecting a solution. |
| Brute-Force with Excessive Complexity | Large language models attempt to solve problems through brute force search, but the search space is too large, making it difficult to find a solution. |
| Factual/Hallucinatory | In the intermediate steps of reasoning, large language models fabricate non-existent facts, data, or logical relationships, leading to erroneous conclusions. |
| Over Verification | The correct answer appeared during the reasoning process, but was not ultimately obtained. |
| Infinite Repetition | The model keeps repeating a certain segment during reasoning, resulting in the inability to obtain results or output answers in the specified format. |

Table 9: Error types in UP error analysis.

| Category | Definition |
|---|---|
| Erroneous reasoning | The model genuinely believes, through reasoning, that there is a solution to the problem. |
| Mandatory response | The model did not obtain an effective solution through logical reasoning, but was forced to answer that the problem had a solution in the end. |
| Unable to deduce | The model cannot derive an answer within the maximum token limit (whether or not it has deduced that the problem is unsolvable halfway through). |
| Infinite repetition | The model keeps repeating a certain segment during reasoning, resulting in the inability to obtain results or output answers in the specified format. |

## D    ADDITIONAL ANALYSIS

### D.1    CORRELATION BETWEEN COMPLEXITY AND MODEL ACCURACY

In Section 2.3, we analyze the complexity of HardcoreLogic from an algorithmic perspective. For IC1, we quantify difficulty through the expansion of the search space; for IC2 and UE2, we evaluate solver-level metrics such as conflicts, decisions, generated nodes, and expanded nodes. These indicators provide a principled way to assess puzzle hardness under classical algorithmic or constraint-solving paradigms.

However, whether these transformations indeed increase difficulty for LRMs remains an empirical question. To align algorithmic hardness with LRM performance, we conduct an additional analysis in this part

For each puzzle instance, our evaluation adopts an $n_{sampling} = 4$ protocol, where a model is queried four times and the instance-level success rate is computed as the proportion of error-free outputs. To examine how solver-based complexity measures relate to LRM performance, we correlate these success rates with classical complexity.

Figure 12, Figure 13, and Figure 14 summarize how LRM success rates vary with different complexity indicators under IC1, IC2, and UE2. To quantitatively validate these relationships, Table 10- 19 report the corresponding significance tests, showing the statistical strength of these complexity–performance correlations across all models.

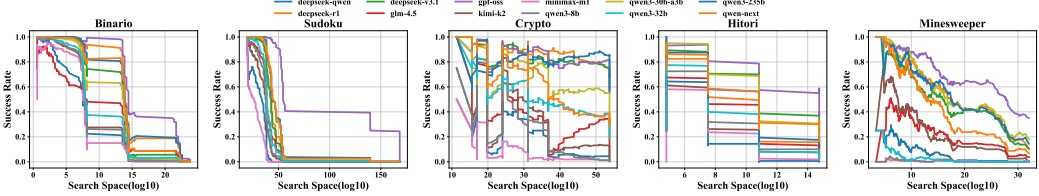

Figure 12: Correlation between IC1 complexity indicators and LRM success rates

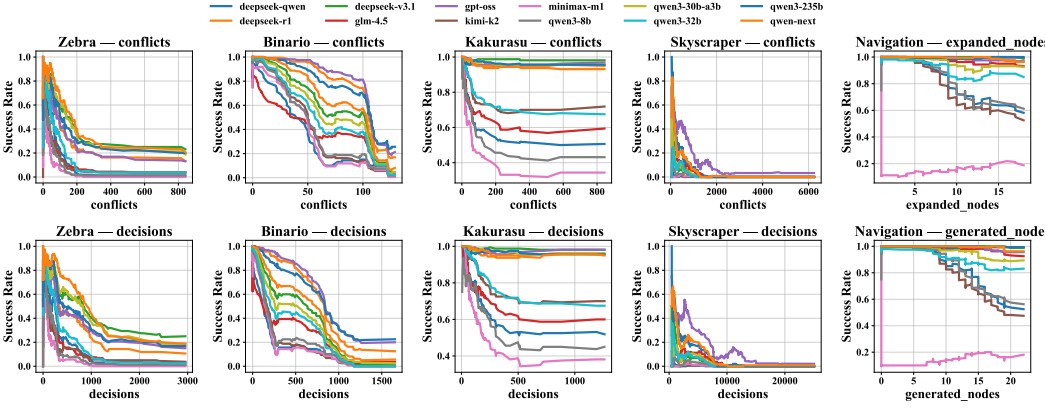

Figure 13: Correlation between solver-based IC2 complexity indicators and LRM success rates.

### D.2    KEY CELLS VS. COMPLEXITY

Among our 10 puzzles, the Search puzzles include Hitori, Minesweeper, and Kakurasu. They all have one thing in common: searching for (or deleting) certain key cells. However, we observe that increasing these key cells does not necessarily make the puzzle harder:

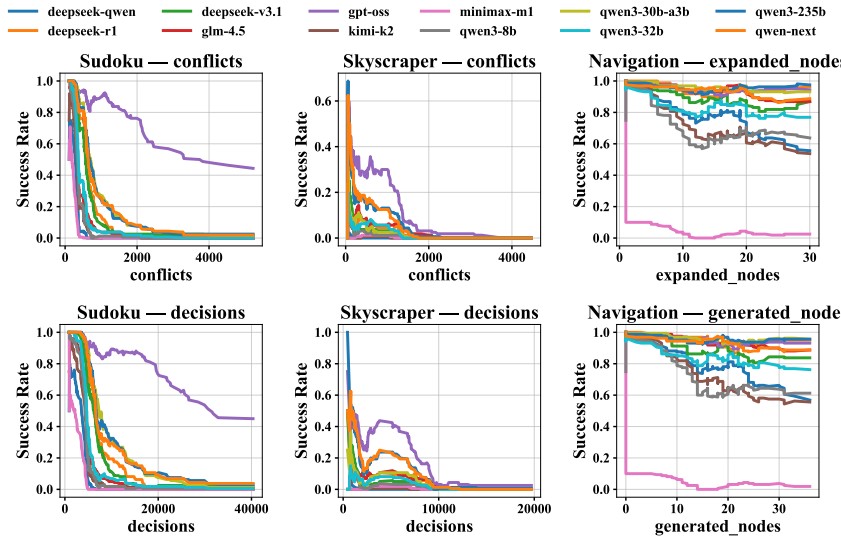

Figure 14: Correlation between UE2 complexity indicators and LRM success rates.

Table 10: P-values from significance tests evaluating the relationship between IC1 **search space** complexity and LRM success rates (Part 1).

| Game | deepseek qwen | deepseek r1 | deepseek v3.1 | glm 4.5 | gpt-oss-120b | kimi-k2 |
|---|---|---|---|---|---|---|
| Binario | 4.24e-83 | 2.21e-119 | 1.35e-114 | 2.45e-81 | 1.88e-106 | 6.73e-96 |
| Crypto | 3.20e-15 | 3.32e-04 | 2.22e-17 | 1.16e-37 | 1.64e-16 | 1.82e-50 |
| Hitori | 6.25e-29 | 5.58e-35 | 1.51e-25 | 8.19e-24 | 1.69e-19 | 4.81e-29 |
| Minesweeper | 3.95e-10 | 1.79e-35 | 2.94e-32 | 3.41e-12 | 1.05e-36 | 5.71e-20 |
| Sudoku | 2.97e-09 | 3.64e-22 | 7.88e-22 | 5.66e-17 | 2.46e-54 | 1.23e-13 |

Table 11: P-values from significance tests evaluating the relationship between IC1 **search space** complexity and LRM success rates (Part 2).

| Game | minimax-m1 | qwen3-235b | qwen3-30b-a3b | qwen3-32b | qwen3-8b | qwen-next |
|---|---|---|---|---|---|---|
| Binario | 3.99e-75 | 6.76e-111 | 1.32e-111 | 4.82e-101 | 1.02e-91 | 6.90e-113 |
| Crypto | 2.18e-10 | 2.16e-07 | 1.63e-31 | 4.90e-32 | 1.35e-39 | 6.58e-35 |
| Hitori | 1.39e-27 | 2.35e-34 | 7.54e-36 | 8.29e-37 | 2.20e-32 | 1.01e-27 |
| Minesweeper | 5.34e-01 | 1.10e-32 | 1.12e-34 | 4.20e-05 | 4.05e-02 | 3.45e-31 |
| Sudoku | 3.67e-07 | 7.08e-27 | 2.77e-27 | 1.14e-16 | 4.58e-15 | 4.52e-27 |

Table 12: P-values from significance tests evaluating the relationship between IC2 **conflicts** complexity and LRM success rates (Part 1). *For the **Navigation** task, the solver metric used is **expanded nodes***

| Game | deepseek qwen | deepseek r1 | deepseek v3.1 | glm 4.5 | gpt-oss-120b | kimi-k2 |
|---|---|---|---|---|---|---|
| Binario | 5.13e-51 | 3.84e-50 | 4.86e-60 | 2.36e-41 | 5.29e-49 | 1.67e-68 |
| Kakurasu | 8.01e-03 | 1.78e-01 | 7.90e-01 | 4.55e-01 | 5.34e-01 | 5.01e-01 |
| Skyscraper | 2.24e-01 | 7.67e-07 | 1.77e-06 | 2.55e-08 | 2.43e-16 | 2.78e-01 |
| Zebra | 1.53e-16 | 1.00e-21 | 1.41e-14 | 1.74e-13 | 6.88e-23 | 2.15e-08 |
| Navigation | 7.38e-20 | 1.55e-01 | 7.53e-03 | 1.82e-06 | 2.28e-04 | 5.65e-26 |

Table 13: P-values from significance tests evaluating the relationship between IC2 **conflicts** complexity and LRM success rates (Part 2). *For the **Navigation** task, the solver metric used is **expanded nodes***

| Game | minimax-m1 | qwen3-235b | qwen3-30b-a3b | qwen3-32b | qwen3-8b | qwen-next |
|---|---|---|---|---|---|---|
| Binario | 1.71e-53 | 5.88e-47 | 7.36e-64 | 8.48e-58 | 7.98e-59 | 4.73e-61 |
| Kakurasu | 1.66e-01 | 5.58e-01 | 7.72e-01 | 5.38e-05 | 2.28e-02 | 1.51e-01 |
| Skyscraper | 9.27e-02 | 6.08e-15 | 9.09e-07 | 3.70e-07 | 2.31e-02 | 9.18e-14 |
| Zebra | 9.80e-09 | 9.10e-21 | 1.65e-20 | 2.71e-17 | 1.85e-12 | 2.25e-23 |
| Navigation | 1.16e-01 | 3.56e-01 | 3.74e-04 | 2.70e-07 | 4.37e-23 | 3.25e-04 |

Table 14: P-values from significance tests evaluating the relationship between IC2 **decisions** complexity and LRM success rates (Part 1). *For the **Navigation** task, the solver metric used is **generated nodes***

| Game | deepseek qwen | deepseek r1 | deepseek v3.1 | glm 4.5 | gpt-oss-120b | kimi-k2 |
|---|---|---|---|---|---|---|
| Binario | 2.35e-32 | 4.02e-63 | 1.76e-56 | 6.37e-35 | 1.03e-64 | 2.79e-41 |
| Kakurasu | 1.29e-03 | 3.00e-01 | 8.34e-01 | 1.55e-01 | 5.27e-01 | 1.71e-01 |
| Skyscraper | 2.11e-01 | 1.58e-07 | 1.49e-07 | 1.13e-09 | 6.72e-19 | 2.12e-01 |
| Zebra | 3.62e-22 | 4.10e-29 | 9.19e-18 | 4.51e-18 | 3.54e-28 | 7.82e-11 |
| Navigation | 6.69e-22 | 1.30e-01 | 3.41e-02 | 9.42e-07 | 1.01e-04 | 2.45e-28 |

Table 15: P-values from significance tests evaluating the relationship between UE2 **decisions** complexity and LRM success rates (Part 2). *For the **Navigation** task, the solver metric used is **generated nodes***

| Game | minimax-m1 | qwen3-235b | qwen3-30b-a3b | qwen3-32b | qwen3-8b | qwen-next |
|---|---|---|---|---|---|---|
| Binario | 6.33e-34 | 6.30e-57 | 2.10e-53 | 6.56e-47 | 7.50e-38 | 3.77e-61 |
| Kakurasu | 6.17e-02 | 4.03e-01 | 5.57e-01 | 1.29e-05 | 1.11e-02 | 3.27e-02 |
| Skyscraper | 6.59e-02 | 1.11e-17 | 1.25e-07 | 1.63e-08 | 1.21e-02 | 7.64e-16 |
| Zebra | 3.59e-11 | 2.03e-27 | 2.51e-27 | 2.94e-23 | 3.67e-16 | 5.83e-32 |
| Navigation | 1.13e-01 | 1.83e-01 | 3.38e-05 | 6.02e-07 | 6.15e-25 | 9.91e-04 |

Table 16: P-values from significance tests evaluating the relationship between UE2 **conflicts** complexity and LRM success rates (Part 1). *For the **Navigation** task, the solver metric used is **expanded nodes***

| Game | deepseek qwen | deepseek r1 | deepseek v3.1 | glm 4.5 | gpt-oss-120b | kimi-k2 |
|---|---|---|---|---|---|---|
| Skyscraper | 3.77e-01 | 1.20e-03 | 4.05e-03 | 3.43e-04 | 3.33e-10 | 5.60e-01 |
| Sudoku | 5.74e-08 | 4.48e-27 | 1.23e-24 | 1.13e-14 | 1.60e-38 | 5.49e-12 |
| Navigation | 2.60e-15 | 2.98e-02 | 1.47e-04 | 1.51e-06 | 1.79e-03 | 3.57e-18 |

Table 17: P-values from significance tests evaluating the relationship between UE2 **conflicts** complexity and LRM success rates (Part 2). *For the **Navigation** task, the solver metric used is **expanded nodes***

| Game | minimax-m1 | qwen3-235b | qwen3-30b-a3b | qwen3-32b | qwen3-8b | qwen-next |
|---|---|---|---|---|---|---|
| Skyscraper | 2.98e-01 | 2.99e-07 | 1.12e-03 | 7.90e-04 | 5.36e-02 | 4.92e-09 |
| Sudoku | 2.88e-06 | 6.85e-31 | 4.06e-33 | 1.52e-15 | 1.32e-12 | 1.56e-29 |
| Navigation | 6.89e-09 | 6.79e-02 | 2.56e-03 | 2.29e-05 | 1.89e-14 | 1.78e-04 |

Table 18: P-values from significance tests evaluating the relationship between UE2 **decisions** complexity and LRM success rates (Part 1). *For the **Navigation** task, the solver metric used is **generated nodes***

| Game | deepseek qwen | deepseek r1 | deepseek v3.1 | glm 4.5 | gpt-oss-120b | kimi-k2 |
|---|---|---|---|---|---|---|
| Skyscraper | 2.60e-01 | 2.16e-04 | 7.46e-04 | 2.96e-05 | 2.27e-13 | 4.65e-01 |
| Sudoku | 1.61e-10 | 2.42e-33 | 4.10e-30 | 6.54e-19 | 1.53e-37 | 2.16e-15 |
| Navigation | 7.58e-17 | 8.82e-03 | 1.87e-04 | 8.31e-07 | 5.55e-04 | 4.99e-21 |

Table 19: P-values from significance tests evaluating the relationship between UE2 **decisions** complexity and LRM success rates (Part 2). *For the **Navigation** task, the solver metric used is **generated nodes***

| Game | minimax-m1 | qwen3-235b | qwen3-30b-a3b | qwen3-32b | qwen3-8b | qwen-next |
|---|---|---|---|---|---|---|
| Skyscraper | 2.19e-01 | 1.31e-09 | 2.48e-04 | 3.61e-05 | 2.11e-02 | 4.64e-12 |
| Sudoku | 3.14e-08 | 1.47e-37 | 3.61e-41 | 1.48e-19 | 2.83e-16 | 2.91e-36 |
| Navigation | 2.02e-09 | 3.49e-02 | 9.99e-04 | 5.25e-06 | 1.13e-18 | 3.83e-04 |

- Under the same grid size, increasing the number of cells to be erased in Hitori does not make it more difficult according to the CSP solver.
- Minesweeper from Original are leveled according to the number of landmines; however, the search space does not vary much.
- On Kakurasu, increasing the number of marked cells also increases the conflicts of decisions, which is the sole positive case.

We tested the performance of Hitori when only increasing the number of cells to be searched(results shown in Table 20), and found that there was no significant difference in performance compared to the Original data when the model was large, but there was a significant difference when the model was small. The grading of Minesweeper also indicates this conclusion that increasing the number of cells to be searched is more difficult for smaller models. For models with insufficient reasoning ability, it is not possible to think about multiple cells in a mixed manner, and it is necessary to think about each cell. Whenever they determine whether a cell is the one they need to find, the probability of errors increases, and increasing the number of cells that need to be found makes it difficult. Due to the unclear impact of this factor, we did not consider it as an independent long-tail transformation. HardcoreLogic has an average of more cells to find for on search puzzles of the same size than the Original dataset.

Table 20: Performance on Hitori of the same size. Compared with the data from Original, the data from HardcoreLogic requires more cells to be searched.

| Data type | gpt-oss-120b | qwen3-235b | qwen3-8b |
|---|---|---|---|
| Original-$4 \times 4$ | 91.00 | 88.00 | 68.00 |
| Original-$5 \times 5$ | 81.50 | 55.00 | 29.50 |
| HardcoreLogic-$4 \times 4$ | 90.50 | 85.50 | 62.50 |
| HardcoreLogic-$5 \times 5$ | 84.00 | 47.00 | 21.50 |

### D.3 ERROR TYPE ANNOTATION CONSISTENCY ANALYSIS

In Sections 4.2 and 4.3, we conducted detailed error analyses for both regular reasoning failures and UP cases, covering UP-error and UP-sufficient categories. For all sampled instances, the final labels were obtained through a voting-based annotation scheme involving three annotator LLMs (Gemini-2.5 Pro, Claude Sonnet 4.5, and GPT-5), followed by manual resolution when no majority vote was reached. Table 21 and Table 22 report the consistency analysis of these annotations. We use Fleiss' Kappa to measure agreement among the three annotator models, and Cohen's Kappa to quantify the agreement between each individual annotator and the final (three LLMs-human hybrid)

labels. The results show generally high agreement, especially the consistently strong alignment between GPT-5 and the final annotations, indicating the reliability of the labeling process.

Table 21: Inter-annotator agreement for error-type labels across three annotator LLMs (Gemini, Claude, GPT-5) on both Original and HardcoreLogic datasets. The table reports **Fleiss' Kappa** for multi-rater agreement and pairwise **Cohen's Kappa** between each annotator and the final voted label.

|  |  | gpt-oss-120b | kimi-k2 | minimax-m1 | qwen3-235b | overall |
|---|---|---|---|---|---|---|
| **Original** | Gemini–Claude–GPT5 | 0.32 | 0.51 | 0.55 | 0.36 | 0.50 |
|  | Gemini vs final | 0.35 | 0.75 | 0.78 | 0.55 | 0.64 |
|  | Claude vs final | 0.63 | 0.54 | 0.87 | 0.54 | 0.68 |
|  | GPT5 vs final | 0.82 | 0.87 | 0.59 | 0.83 | 0.81 |
| **Hardcore** | Gemini–Claude–GPT5 | 0.37 | 0.29 | 0.52 | 0.34 | 0.43 |
|  | Gemini vs final | 0.50 | 0.62 | 0.74 | 0.66 | 0.66 |
|  | Claude vs final | 0.51 | 0.38 | 0.77 | 0.34 | 0.54 |
|  | GPT5 vs final | 0.92 | 0.67 | 0.72 | 0.87 | 0.81 |
| **Both** | Gemini–Claude–GPT5 | 0.36 | 0.41 | 0.55 | 0.36 | 0.47 |
|  | Gemini vs final | 0.43 | 0.69 | 0.76 | 0.61 | 0.65 |
|  | Claude vs final | 0.57 | 0.46 | 0.82 | 0.44 | 0.61 |
|  | GPT5 vs final | 0.87 | 0.78 | 0.67 | 0.85 | 0.81 |

Table 22: Inter-annotator agreement for **UP-error** and **UP-sufficient** cases. Similar to Table 21, the table includes **Fleiss' Kappa** across the three annotator LLMs and pairwise **Cohen's Kappa** with the final voted label, reflecting the reliability of annotations in the unsolvable-puzzle setting.

|  |  | gpt-oss-120b | kimi-k2 | minimax-m1 | qwen3-235b | overall |
|---|---|---|---|---|---|---|
| **error** | Gemini–Claude–GPT5 | 0.55 | 0.25 | 0.45 | 0.35 | 0.54 |
|  | Gemini vs final | 0.48 | 0.58 | 0.65 | 0.43 | 0.62 |
|  | Claude vs final | 0.88 | 0.21 | 0.48 | 0.60 | 0.69 |
|  | GPT5 vs final | 1.00 | 0.91 | 0.92 | 0.93 | 0.96 |
| **sufficient** | Gemini–Claude–GPT5 | 0.51 | 0.36 | 0.019 | -0.02 | 0.27 |
|  | Gemini vs final | 0.54 | 1.00 | 0.00 | 0.00 | 0.39 |
|  | Claude vs final | 1.00 | 0.65 | 0.37 | 0.00 | 0.58 |
|  | GPT5 vs final | 0.67 | 0.43 | 0.71 | 0.85 | 0.66 |

## D.4 SKYSCRAPER SOLUTION COUNT

We found that almost all models performed poorly in solving Skyscraper, due to the difficulty of the problem itself. We found that the number of solutions to such difficult puzzles may affect the performance of the model. We performed two different long-tail transformations on Skyscraper: add diagonal constraints and hide partial clues. These two types of long-tail transformations are referred to as diag and partial. These two types of long-tail transformations show improvements in both decisions and conflicts compared to the Original dataset at the same size. However, we found that on some well-performing models, the accuracy of large-sized ($6 \times 6$ and above) partial transformations (without guaranteed unique solutions) partially increased, while diagonal transformations and $5 \times 5$ partial transformations (with guaranteed unique solutions) showed a significant downward trend in model performance(results shown in Table 23). Large-sized partial transformations result in an increase in the number of solutions due to hidden clues, which affects the performance of the model. The partial transformation and $5 \times 5$ diagonal transformation ensure that the solution does not increase compared to the Original dataset, and with the increase of decisions and conflicts, even in some diagonal transformation data that can be solved with the original constraints, the performance of the model still decreases significantly. So when the puzzle is difficult and the model does not have enough clues to analyze, it may tend to guess the answer, and the number of solutions becomes a factor affecting the difficulty of the game.

Table 23: The performance of some models on Skyscraper with sizes of $5 \times 5$ and $6 \times 6$, using the count hint.

| Data type | gpt-oss-120b | deepseek-v3.1 | qwen3-235b |
|---|---|---|---|
| Original-$5 \times 5$ | 41.30 | 14.13 | 30.43 |
| Original-$6 \times 6$ | 1.85 | 0.00 | 0.00 |
| HardcoreLogic-diag-$5 \times 6$ | 19.50 | 1.50 | 8.50 |
| HardcoreLogic-diag-$6 \times 6$ | 0.50 | 0.00 | 0.00 |
| HardcoreLogic-partial-$5 \times 5$ | 24.50 | 7.00 | 22.00 |
| HardcoreLogic-partial-$6 \times 6$ | 5.00 | 0.00 | 0.00 |

## D.5   OTHER ANALYSES OF WEIGHTED MULTIPLE LINEAR REGRESSION

In Section 4.1, we performed weighted multiple linear regression to examine the effects of four different long-tail transformations on puzzle difficulty. Concretely, we fit the following model:

$$y = k_{\text{IC1}} \cdot 1_{\text{IC1}} + k_{\text{IC2}} \cdot 1_{\text{IC2}} + k_{\text{UE1}} \cdot 1_{\text{UE1}} + k_{\text{UE2}} \cdot 1_{\text{UE2}} + b$$

where $y$ is the observed accuracy for a specific puzzle variant, $1_{\text{IC1}}$ is a binary indicator (1 if transformation IC1 is applied, 0 otherwise), $k_{\text{IC1}}$ quantifies the marginal accuracy change attributable to IC1 under the assumption of additive effects, $b$ is the expected accuracy predicted by the model when all dummy variables are zero, and weights $w_i = N_i$ (sample sizes) give greater influence to observations with larger sample sizes when calculating the loss function. Weighted linear regression isolates the marginal effect of individual transformations through two mechanisms: (1) the additive linear model with dummy variables statistically disentangles combined transformation effects by estimating each factor's contribution relative to the baseline configuration, (2) sample-size-based weighting assigns greater influence to high-reliability observations during coefficient estimation, ensuring parameters reflect dominant patterns in robust data.

To complement the results presented in Section 4.1 of the main text, this appendix provides additional details of the weighted multiple linear regression analysis. First, we refitted the model using data that contained only a factor, excluding all data points that included multiple factors. Second, based on the original multivariate model, we computed the corresponding 95% confidence intervals (and corresponding p-values) of the regression coefficients.

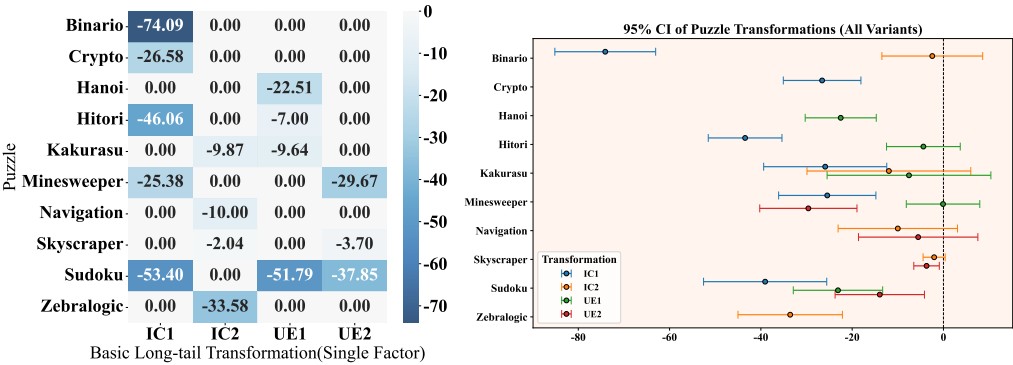

Figure 15:  **Left**: Effects of long-tail transformations on puzzle accuracy.(single factor) **Right**: 95% confidence intervals for puzzle difficulty coefficients.

The left side of Figure 15 shows the impact of long-tail transformations on puzzle accuracy when only considering single factor data. For most puzzles, the coefficients obtained from this simulation closely match those from the full multiple regression model. The only notable deviation occurs in Sudoku. This is because the UE1 category for Sudoku actually contains two heterogeneous subtypes—letter version and irregular subgrid. The letter version variant has only a minor standalone effect and appears only in combination with other variants, whereas the irregular subgrid variant never co-occurs

Table 24: $p$-values of the fitted weighted linear regression.

| Puzzle | IC1 | IC2 | UE1 | UE2 | Puzzle | IC1 | IC2 | UE1 | UE2 |
|---|---|---|---|---|---|---|---|---|---|
| **ZebraLogic** | — | .000 | — | — | **Minesweeper** | .000 | — | .982 | .000 |
| **Sudoku** | .000 | — | .000 | .006 | **Navigation** | — | .132 | — | .402 |
| **Skyscraper** | — | .098 | — | .010 | **Binario** | .000 | .661 | — | — |
| **Kakurasu** | .000 | .188 | .403 | — | **Hanoi** | — | — | .000 | — |
| **Crypto** | .000 | — | — | — | **Hitori** | .000 | — | .283 | — |

with any other factors. As a result, when simulating Sudoku using single-factor data, the model's intercept becomes shifted, which in turn leads to changes in the estimated parameters.

The right panel of Figure 15 presents the 95% confidence intervals of the puzzle-difficulty coefficients, with the corresponding p-values reported in Table 24 . Most of the confidence interval bounds are negative, and the overall conclusions are consistent with those in Section 4.1. The figure further shows that, even after accounting for estimation uncertainty, IC1 still exhibits the largest effect size in our data. Moreover, all p-values associated with IC1 are below 0.001, confirming that its influence on puzzle difficulty is statistically significant.

