# OpenReview forum: "HardcoreLogic: Challenging Large Reasoning Models with Long-tail Logic Puzzle Games"
_ICLR.cc/2026/Conference — ICLR 2026 Poster_

### Official Review · Reviewer_eXAq · 2025-10-24

**Soundness:** 3
**Presentation:** 3
**Contribution:** 3
**Rating:** 8
**Confidence:** 4

**Summary:**

In this paper, the authors introduce HardcoreLogic, a new benchmark of puzzle and algorithmic reasoning problems that systematically expands the complexity of several well-known puzzle datasets (e.g., Zebralogic, Enigmata, Hanoi game and the others shown in Table 1). In particular, they use the taxonomy of transformations detailed in Section 2.2, which includes increasing problem complexity (via increases to the search space and depth of reasoning), constraint strengthening, modificaitons to question forms, rule variation, and introduce an unsolvable puzzle setting that are not (as I understand) native to any of the existing datasets.

Given that most datasets are carefully contructed using algorithmic tools like Z3, creating such modifications is not an altogether straightforward task. Figure 2 shows the increase in the search space size of the resulting puzzles, and Figure 3 very nicely illustrates the relative differences (in the green and purple plots) in the numbers of SAT conflicts and decisions (often a proxy for empirical hardness of a problem) between the old and new datasets. Computationally, these problems are indeed considerably more difficult (I'm naturally wondering, however, how such added complexity increases the raw size of the input problem. Perhaps I missed this detail somewhere?)

They experiment with a comprehensive set of both commerical and open weight large reasoning models (GPT-5, GPT-oss, deepseek, qwen3, etc.) and a single non-reasoning model kimi-k2. Figure 4 shows the aggregate performance of different models across old and new puzzle tasks, where all models suffer significant reductions in performance (some of the particular details here are interesting, such as the relative robustness of the GPT models and Grok). Dataset-level results are shown in Figure 5, which does indicate that the degree of difference is quite mixed across these different tasks.

To better understand the effects of the particular puzzle transformations on model performance, they fit a linear regression to each puzzle task, the results of which are reported in Figure 6. I find this analysis to be very interesting and unique, as it could provide a recipe for how to create an even harder set of tasks.  They also report some interesting error analysis.

**Strengths:**

-- A **new set of hard puzzle datasets** that systematically improve the complexity and difficulty of several well benchmarks. I could imagine this dataset being the new standard resource for these class of problems.

-- **Compelling empirical evidence** about the hardness of the tasks, with very interesting analysis concerning the factors that contribute to the increased difficulty and current error cases.

-- A very clear and easy to read paper that will be approachable for those not working directly in this area.

**Weaknesses:**

-- (**minor**) Whenever complexity is increased, especially for transformations involving novel rules (i.e., their *uncommon elements*), one runs the risk of introducing ambiguous or impossible to understand constraints. I'd be curious to know if anything was done to mitigate such issues (e.g., manual analysis of some kind or human evaluation) and would like to have the authors address this.

**Questions:**

-- Did you experiment any anthropic models?

-- My question from above: how does the increase complexity affect the input size of each problem? I worry that making problems too complex might result in puzzles that are very cumbersome and far too verbose to be realistic.

-- Why did the transformations have such a small effect on the skyscaper tasks (reported in Figure 5)?

---

> ### Author Response · Authors · 2025-11-21
>
> ## A1: Transformation comprehensibility
>
> > **W1 & Q2:** Whenever complexity is increased, especially for transformations involving novel rules (i.e., their uncommon elements), one runs the risk of introducing ambiguous or impossible to understand constraints. I'd be curious to know if anything was done to mitigate such issues (e.g., manual analysis of some kind or human evaluation) and would like to have the authors address this.
>
> Thank you for raising this concern. We take several steps to mitigate the risk of introducing ambiguous or unintelligible constraints.
>
> First, during the transformation design stage, we ensure that **every transformation is fully compatible with rule-based validation**, guaranteeing that all transformed puzzles retain well-defined constraints.
>
> Second, for the prompt format, we conduct **manual testing across multiple model families**, iterating over several prompt versions and selecting the format that consistently results in the least misunderstanding. This helps ensure that the final prompts remain clear and interpretable across different reasoning models.
>
> Third, regarding **prompt complexity**, all HardcoreLogic prompts remain under $1,000$ **tokens**, except for 16x16 Sudoku puzzles that naturally require around $1,600$ tokens due to the large board description. These prompt lengths are still within the moderate range, especially compared to other grid-based benchmarks like ARC-AGI-2 [1] that require significantly longer inputs.
>
> Finally, **Table R1** compares prompt lengths for puzzles of similar scale in the **Original v.s. HardcoreLogic datasets**. The results show that HardcoreLogic prompts are not substantially longer than their Original counterparts (except for Navigation because all puzzles in HardcoreLogic are significantly larger than the ones in Original). This demonstrates that the added complexity from long-tail transformations does not inherently make the prompts more difficult to understand.
>
> We will incorporate these clarifications in the revised manuscript to address the reviewer's concerns.
>
> | Game     | Original | HardcoreLogic | Game          |          Original |        HardcoreLogic |
> | :------- | -------: | ------------: | :------------ | ----------------: | -------------------: |
> | Binario  | $350.51$ |     $+2.04\\%$ | Minesweeper   |          $403.58$ |            $+7.26\\%$ |
> | Crypto   | $310.57$ |     $+2.38\\%$ | _Navigation*_ | $\mathit{287.03}$ | $\mathit{+139.08\\%}$ |
> | Hanoi    | $265.25$ |     $+2.18\\%$ | Skyscraper    |          $613.50$ |            $-0.45\\%$ |
> | Hitori   | $297.00$ |    $+11.20\\%$ | Sudoku        |          $722.54$ |            $+1.31\\%$ |
> | Kakurasu | $452.20$ |     $-0.08\\%$ | ZebraLogic    |          $700.66$ |           $+13.80\\%$ |
>
> > **Table R1:** Prompt length difference between Original and HardcoreLogic on puzzles with similar scale (except for *Navigation* where HardcoreLogic puzzles are all larger than Baseline ones). Original columns show the average prompt length of each game, while HardcoreLogic columns show the difference ratio.
>
> [1] Chollet et. al., *ARC-AGI-2: A New Challenge for Frontier AI Reasoning Systems.*
>
> ## A2: Anthropic models
>
> > **Q1:** Did you experiment any anthropic models?
>
> Yes, as indicated in Figure 4 (Section 3.2) and Table 7 (Appendix C.1), we have evaluated **Claude Sonnet 4** from Anthropic and reported the results.

---

> > ### Author Response · Authors · 2025-11-21
> >
> > ## A3: Skyscraper results
> >
> > > **Q3:** Why did the transformations have such a small effect on the skyscaper tasks (reported in Figure 5)?
> >
> > Thank you for the question. Skyscraper is the only game in HardcoreLogic whose canonical puzzles in Original **are already highly difficult**, leaving relatively little room for additional degradation when transformations are applied. We believe Skyscraper is intrinsically challenging for LRMs mainly for two reasons:
> >
> > 1. Its visibility rules require **non-linear**, state-dependent reasoning that LLMs struggle to simulate.
> > 2. the puzzle offers only **weak local heuristics** where most placements do not immediately reveal constraint violations, making it difficult for models to iteratively prune invalid states.
> >
> > Furthermore, we observe that even after doubling the reasoning-length budget from $32,768$ to $65,536$, all models show only slight improvement but still remain below 20% accuracy (see **Table R2**). This confirms that the canonical Skyscraper puzzles are already extremely challenging.
> >
> > | Benchmark     | gpt-oss-120b | (65,536) | seed-oss-36b | (65,536) |
> > | :------------ | -----------: | -------: | -----------: | -------: |
> > | Original      |      $15.50$ |  $17.12$ |      $10.62$ |  $17.25$ |
> > | HardcoreLogic |       $9.04$ |   $9.38$ |       $4.71$ |   $7.00$ |
> > | **Gap**       |      $-6.46$ |  $-7.74$ |      $-5.91$ | $-10.25$ |
> >
> > > **Table R2:** Model accuracy (%) of **gpt-oss-120b** and **seed-oss-36b** on Skyscraper under two reasoning-length budgets ($32,768$ on model-name columns, $65,536$ tokens on columns to their right). We report results on both Original and HardcoreLogic, as well as the gap between them.
> >
> > Given that the canonical Skyscraper puzzles are already very hard, we deliberately avoid injecting excessive long-tail transformations into this game, as doing so would reduce its ability to differentiate model performance. Moreover, despite this high baseline difficulty, our transformations still cause over a $40\\%$ relative accuracy drop for all models.
> >
> > We have added clarifications and updated results in the revised manuscript.

---

> > > ### Comment · Reviewer_eXAq · 2025-11-25
> > >
> > > Thank you for your response. I particular found these details about *Transformation comprehensibility* to be reassuring. Since my score was already high, I will keept it as it.

---

### Official Review · Reviewer_xWwM · 2025-10-26

**Soundness:** 3
**Presentation:** 2
**Contribution:** 2
**Rating:** 4
**Confidence:** 3

**Summary:**

This work introduce HardcoreLogic, a challenging benchmark comprising over 5,000 long-tail variants of 10 logical puzzle games, designed to test the robustness and genuine high-level logical reasoning of Large Reasoning Models . To bypass model reliance on memorized canonical patterns, the benchmark systematically transforms puzzles using Increased Complexity, Uncommon Elements, and Unsolvable Puzzles. Evaluations on a diverse set of LRMs, including SOTA models like GPT-5, revealed a significant performance degradation, indicating that current models struggle with less conventional scenarios and often rely on memorization rather than flexible reasoning. The dominant source of difficulty is the Increased Complexity, which tests the model’s ability to handle larger search spaces and greater constraint entanglement.

**Strengths:**

1. The work introduce a challenging benchmark, which helps evaluate the reasoning and generalization capabilities of LRMs.
2. The statistics and the analysis on the model responses are informative, which reveals the weaknesses of the LRMs.

**Weaknesses:**

1. The data construction process is not so clear. It is not clear whether datasets are currates manually, rule-based, or by LLMs. How do you guarantee the validity of the transformed questions and answers.
2. Some important experimental and analysis are also missing. How do you evaluate the correctness of the responses?
3. For the error analysis, GPT-5 was used to do the classification. It is not clear how robust is such setting. Manual analysis may be necessary for more reliable analysis. In section 4.3, are the analysis conducted by humans or LLMs? How many samples do you use for the analysis?
4. One important concern of this work is the novelty. The main contribution of this is a transformed benchmark that is more difficult and diverse but still similar to the original benchmark, which limits its novelty.

**Questions:**

When curating the benchmark, it would be great for show more examples so that the readers can understand the details of the data construction process.

---

> ### Author Response · Authors · 2025-11-21
>
> ## A1: Data construction pipeline
>
> > **W1 & Q1:** The data construction process is not so clear. It is not clear whether datasets are currates manually, rule-based, or by LLMs. How do you guarantee the validity of the transformed questions and answers.
>
> Due to space limitations, the detailed data construction process was placed in Appendix B, but we agree that a clearer explanation is needed. All puzzles except ZebraLogic are generated completely through rule-based programs, and we use deterministical validation programs to ensure that all answers are correct. For ZebraLogic, we adopt a two-stage rule-LLM hybrid procedure to create the natural-language problems as prior benchmarks:
>
> 1. **Rule-based generation and validation:** We first generate each puzzle in a fully symbolic, rule-based form and validate its solution programmatically.
>
> 2. **LLM rewriting for natural-language prompt:** we then use gpt-4o-mini to rewrite the clues in natural language. To ensure correctness and clarity, we apply o4-mini to compare the rewritten clues with the original rule-based version and automatically flag any inconsistencies or ambiguities for correction.
>
> Figure 1 in the original paper illustrates two Sudoku examples with two transformation types applied to the canonical puzzle. Table 3 in Appendix B.2 summarizes the construction details for each game in HardcoreLogic. In the revised manuscript, we further include comprehensive examples for all games in Figure 11 (Appendix B.2) to make the transformation process fully transparent and easier to understand.
>
> ## A2: Output evaluation procedure
>
> > **W2:** How do you evaluate the correctness of the responses?
>
> All responses are judged based on whether their proposed answers are correct through rule-based programs checking. We only concern whether the answer is valid or not, since this standard is objective, fair and easy to implement.
>
> To ensure that the model produces machine-verifiable responses and eliminates any presentation or format mismatch, we explicitly require the mode to return its final prediction in a structured JSON format in the prompt:
>
> ```json
> {"solvable": true/false, "solution": ...}
> ```
>
> Here the solution is a task-specific structured object, whose concrete form varies across puzzle types (e.g., lists of lists for grid-based games, dictionaries for assignments or mappings, nested lists for sequence puzzles, etc.).
>
> We apply regex-based guided decoding (for open-source models) or JSON schema hint (for closed-source APIs, forced to return JSON if possible) to force the model to comply with the desired JSON format. All outputs that do not follow the required JSON format are treated as incorrect (only closed-source models rarely produce non-JSON predictions).
>
> After parsing the model's proposal from the JSON document, for solvable puzzles we use rule-based programs to check whether it reports that the puzzle is solvable and provides a solution that really solves the puzzle; for unsolvable puzzles we simply check whether it returns that the puzzle is unsolvable.

---

> > ### Author Response · Authors · 2025-11-21
> >
> > ## A3: Error analysis
> >
> > > **W3:** For the error analysis, GPT-5 was used to do the classification. It is not clear how robust is such setting. Manual analysis may be necessary for more reliable analysis. In section 4.3, are the analysis conducted by humans or LLMs? How many samples do you use for the analysis?
> >
> > To improve the reliability of error annotations, we extend beyond GPT-5 and incorporate **two additional LLM annotators**, including **Gemini-2.5 Pro** and **Claude Sonnet 4.5**. For each sampled error, we accept the label when **at least two models agree**; otherwise, we perform **manual adjudication**. Across the three LLM annotators, we obtain a **Fleiss'** $\\kappa$ of $0.47$, indicating moderate agreement. **Table R1** reports the updated error analysis using this multi-annotator scheme for both Original and HardcoreLogic datasets. Importantly, the new error distributions closely match those produced by GPT-5 alone: the two annotation sets yield a **Cohen's** $\\kappa$ of $0.81$, reflecting high consistency. As for the error analysis in Section 4.3, the procedure is similar to the one in Section 4.2, and $50$ randomly sampled erroneous cases for each model are used.
> >
> > We have incorporated new annotation schema and updated results in Appendix D.3  of the revised manuscript.
> >
> > | Benchmark     | Error Type          | gpt-oss-120b | Qwen3-235B-A22B | Kimi-K2 | Minimax-M1 |
> > | :------------ | :------------------ | -----------: | --------------: | ------: | ---------: |
> > | Original      | Misunderstanding    |        $2.0$ |           $2.0$ |   $8.0$ |      $2.0$ |
> > |               | Misapplied          |        $2.0$ |           $2.0$ |   $4.0$ |      $0.0$ |
> > |               | Brute Force         |       $20.0$ |          $34.0$ |  $10.0$ |     $22.0$ |
> > |               | Factual Errors      |       $50.0$ |          $38.0$ |  $56.0$ |     $20.0$ |
> > |               | Over Verification   |       $24.0$ |          $24.0$ |  $20.0$ |      $0.0$ |
> > |               | Infinite Repetition |        $2.0$ |           $0.0$ |   $0.0$ |     $56.0$ |
> > | HardcoreLogic | Misunderstanding    |       $10.0$ |          $12.0$ |  $22.0$ |      $2.0$ |
> > |               | Misapplied          |        $8.0$ |          $12.0$ |  $14.0$ |      $2.0$ |
> > |               | Brute Force         |       $28.0$ |          $28.0$ |   $4.0$ |     $32.0$ |
> > |               | Factual Errors      |       $38.0$ |          $34.0$ |  $58.0$ |     $24.0$ |
> > |               | Over Verification   |       $16.0$ |          $12.0$ |   $2.0$ |      $2.0$ |
> > |               | Infinite Repetition |        $0.0$ |           $2.0$ |   $0.0$ |     $38.0$ |
> >
> > > **Table R1:** Error type distribution (%) on $50$ error samples per model, integrating annotations from three annotating LLMs and human adjudication.

---

> > > ### Author Response · Authors · 2025-11-21
> > >
> > > ## A4: Contribution and novelty
> > >
> > > > **W4:** The main contribution of this is a transformed benchmark that is more difficult and diverse but still similar to the original benchmark, which limits its novelty.
> > >
> > > We would like to clarify that our novelty extends far beyond simply releasing a transformed benchmark. Our novelty lies in both **the design of comprehensive long-tail transformations for challenging logic puzzles** and, most importantly, the **detailed and insightful experimental analysis** enabled by this benchmark.
> > >
> > > 1. **Long-tail transformations for logic puzzles meaningfully test genuine reasoning.** A central open question is whether LLMs truly generalize or mainly rely on memorization. In this context, long-tail transformation itself is a significant contribution for reliable evaluation and inspire future improvement by creating out-of-domain variations that reduce memorization benefits. Prior influential works such as GSM-Symbolic [1] and DYVAL [2] demonstrate the value of this direction. Aligning with this idea, HardcoreLogic focuses on logic puzzles as fundamental logical reasoning tasks, can effectively challenge the boundary of reasoning ability and separate it from brute-force memorization.
> > >
> > >    [1] Mirzadeh et. al., *GSM-Symbolic: Understanding the Limitations of Mathematical Reasoning in Large Language Models.*
> > >
> > >    [2] Zhu et. al., *DyVal: Dynamic Evaluation of Large Language Models for Reasoning Tasks.*
> > >
> > > 2. **Logic-puzzle transformations are more challenging than prior question rewriting and composition.** Unlike arithmetic or linear deductive reasoning, challenging logic puzzles require structured and non-linear reasoning. Besides, existing long-tail benchmarks typically rely on lightweight textual perturbations or question composition, whereas we perform comprehensive, rule-grounded transformations across $10$ distinct games, each with unique constraints and solution structures, while rigorously guaranteeing correctness and solution uniqueness. This is a significantly more challenging and under-explored problem.
> > >
> > > 3. **Our data construction pipeline is scalable.** The rule-based generation and validation pipeline can be easily scaled to produce larger training corpora or extended to additional puzzle types. This makes our approach practically valuable not only for evaluation but also for training-time improvements.
> > >
> > > 4. Most importantly, **our work offers extensive and insightful analysis.** Beyond dataset construction, the core novelty of our work lies in deep and systematic experimental investigation, including the effects of different transformation types, error modes on solvable puzzles and reasoning patterns on unsolvable puzzles.  These analyses expose both general and model-specific weaknesses and provide insightful suggestions for future improvements about avoiding brute-force loops and encouraging diverse exploration.

---

### Official Review · Reviewer_izsk · 2025-10-30

**Soundness:** 3
**Presentation:** 3
**Contribution:** 3
**Rating:** 8
**Confidence:** 4

**Summary:**

The paper introduces HardcoreLogic, a new benchmark with 5k puzzles spanning over 10 logic puzzle games designed to probe reasoning models on long-tail of logical. The benchmark is constructed by systematically applying long-tail transformations on exiting logical games/puzzles in primarily three ways: Increased Complexity, Uncommon Elements, and Unsolvable Puzzles, and evaluates a broad set of open and closed models. The study reports substantial accuracy drops relative to “Original” datasets, analyses transformation-induced difficulty and provides analysis on error types and model behaviours for solvable/unsolvable puzzle cases.

**Strengths:**

- The paper introduces HardcoreLogic, a valuable new resource for the community. By evaluating on proposed long-tailed transformed logic puzzles with increased difficulty, unknown element, and addition of unsolvable puzzles, the benchmark highlights through their evaluations and error analysis, that a lot of the time the reasoning models rely on their memorised experiences rather than genuine reasoning over the given problem.
- The inclusion of unsolvable instances to evaluate ability of LLMs reason and comprehend insufficiency of information is uncommon and valuable.
- The analysis is thorough, investigating performance across different game types, long-tail transformations and models (open/closed), with insightful error analysis on models and why they perform poorly on HardcoreLogic.
- The paper is well-written, the methodology is sound, and the results are clear and impactful.

**Weaknesses:**

- The qualitative proxies for hardness (e.g., search-space scale, Z3 decisions/conflicts, generated/expanded nodes) are informative, but the paper does not show whether they correlate with what humans perceive hard or LLM error rates. Without such correlations, it is hard to conclude the transformations reliably push puzzles into the “long-tail”.
- Error analysis lacks reliability and confidence reporting. Although GPT-5 is used as an annotator, the resulting error analysis is hard to trust without inter-annotator agreement or multiple human/LLM annotators. The claims of the benchmark can be vastly improved with improving the error analysis and using human experts (or use of multiple LLM-as-judge models), and proving confidence scores.

**Questions:**

- The paper lack head-to-head comparison of error types between the HardcoreLogic and the original problems. Does factual errors increase in HardcoreLogic vs original? Does Brute-force errors exists in LRMs when solving the original problems?
- What is Z3? not defined anywhere in the paper.
- Figure 7 missing label for orange error type.

---

> ### Author Response · Authors · 2025-11-21
>
> ## A1: Complexity metrics and relation to empirical difficulty
>
> > **W1:** How well do your theoretical complexity measures correlate with what humans perceive hard or LLM error rates? Without such correlations, it is hard to conclude the transformations reliably push puzzles into the "long-tail".
>
> The complexity analysis in Section 2.3 introduces several theoretical metrics to quantify how complex a puzzle is from a structural or logical standpoint (e.g., Z3 decisions/conflicts). The purpose of these metrics is to provide an objective, model-agnostic signal of complexity.
>
> We further conduct an additional analysis correlating these theoretical puzzle complexity with LLM performance across multiple models. Specifically, for each game and each transformation type, we plot model accuracy against the corresponding theoretical complexity scores and perform statistical significance testing. These correlation plots are presented in Figures 12-14 of therevised manuscript and we perform statistical significance testing to quantify the strength of these relationships, with detailed p-value results reported in Tables 10–19 (all analysis are included in Appendix D.1). These results generally show that:
>
> 1. **Higher theoretical complexity correlates with lower model accuracy,** and
> 2. **The effect is most pronounced for models with weaker or less robust reasoning ability.**
>
> These findings confirm that our long-tail transformations reliably increase puzzle difficulty in ways that align with both theoretical complexity measures and empirical LLM error patterns.
>
> > **Q2:** What is Z3? It's not defined anywhere in the paper.
>
> Thank you for pointing out the missing definition. In Section 2.3, Z3 refers to the Satisfiability Modulo Theories (SMT) solver developed by Microsoft Research [1]. We use Z3 to compute the number of solver decisions and conflicts required to satisfy the puzzle constraints under different long-tail transformations, allowing us to quantify how much additional logical complexity each transformation introduces.
>
> [1] Leonardo et. al., *Z3: An efficient smt solver.*

---

> > ### Author Response · Authors · 2025-11-21
> >
> > ## A2: Error analysis
> >
> > > **W2 & Q1:** Although GPT-5 is used as an annotator, the resulting error analysis is hard to trust without inter-annotator agreement or multiple human/LLM annotators. The paper lack head-to-head comparison of error types between the HardcoreLogic and the original problems. Do factual errors increase in HardcoreLogic vs original? Does Brute-force errors exists in LRMs when solving the original problems?
> >
> > To improve the reliability of error annotations, we extend beyond GPT-5 and incorporate **two additional LLM annotators**, including **Gemini-2.5 Pro** and **Claude Sonnet 4.5**. For each sampled error, we accept the label when at least two models agree; otherwise, we perform manual adjudication.
> >
> > Across the three LLM annotators, we obtain a **Fleiss'** $\\kappa$ of $0.47$, indicating moderate agreement. **Table R1** reports the updated error analysis using this multi-annotator scheme for both Original and HardcoreLogic datasets. Importantly, the new error distributions closely match those produced by GPT-5 alone: the two annotation sets yield a **Cohen's** $\\kappa$ of $0.81$, reflecting high consistency.
> >
> > Comparing the error type distributions between Original and HardcoreLogic, we find that Brute-force errors appear in both datasets but higher in HardcoreLogic, reflecting the increased difficulty introduced by the long-tail transformations. HardcoreLogic induces more rules misapplied and misunderstanding errors. This distribution shift aligns with the purpose of HardcoreLogic to include non-canonical puzzles that stereotype knowledge or memorized reasoning patterns fails.
> >
> > We have incorporated new annotation schema and updated results in Appendix D.3 of the revised manuscript.
> >
> > | Benchmark     | Error Type          | gpt-oss-120b | Qwen3-235B-A22B | Kimi-K2 | Minimax-M1 |
> > | :------------ | :------------------ | -----------: | --------------: | ------: | ---------: |
> > | Original      | Misunderstanding    |        $2.0$ |           $2.0$ |   $8.0$ |      $2.0$ |
> > |               | Misapplied          |        $2.0$ |           $2.0$ |   $4.0$ |      $0.0$ |
> > |               | Brute Force         |       $20.0$ |          $34.0$ |  $10.0$ |     $22.0$ |
> > |               | Factual Errors      |       $50.0$ |          $38.0$ |  $56.0$ |     $20.0$ |
> > |               | Over Verification   |       $24.0$ |          $24.0$ |  $20.0$ |      $0.0$ |
> > |               | Infinite Repetition |        $2.0$ |           $0.0$ |   $0.0$ |     $56.0$ |
> > | HardcoreLogic | Misunderstanding    |       $10.0$ |          $12.0$ |  $22.0$ |      $2.0$ |
> > |               | Misapplied          |        $8.0$ |          $12.0$ |  $14.0$ |      $2.0$ |
> > |               | Brute Force         |       $28.0$ |          $28.0$ |   $4.0$ |     $32.0$ |
> > |               | Factual Errors      |       $38.0$ |          $34.0$ |  $58.0$ |     $24.0$ |
> > |               | Over Verification   |       $16.0$ |          $12.0$ |   $2.0$ |      $2.0$ |
> > |               | Infinite Repetition |        $0.0$ |           $2.0$ |   $0.0$ |     $38.0$ |
> >
> > > **Table R1:** Error type distribution (%) on $50$ error samples per model, integrating annotations from three annotating LLMs and human adjudication.
> >
> > ## A3: Missing label in Figure 7
> >
> > > **Q3:** Figure 7 missing label for orange error type.
> >
> > Thank you for pointing out the incorrect legend in Figure 7. The orange pie refers to Infinite Repetition (the last type in the legend); we will fix the legend in the revised manuscript.

---

### Official Review · Reviewer_na2W · 2025-11-01

**Soundness:** 2
**Presentation:** 2
**Contribution:** 2
**Rating:** 4
**Confidence:** 3

**Summary:**

This paper builds a new benchmark, HardcoreLogic, to test whether large reasoning models (LRMs) can truly reason or are just memorizing common puzzle patterns. The authors build a dataset of over 5,000 puzzles across 10 different games such as Sudoku, ZebraLogic, and Minesweeper. These are drawn from or aligned with earlier logic-reasoning datasets (like Enigmata, ZebraLogic), but then systematically transformed. Unlike existing datasets, HardcoreLogic includes three types of harder variants: Increased Complexity (IC) – bigger puzzles or denser constraints; Uncommon Elements (UE) – unusual formats or new rules, and Unsolvable Puzzles (UP) – puzzles with no valid answer. When tested on this benchmark, even top-performing models that excel on standard puzzles showed large drops in accuracy, especially when the puzzles were slightly altered or unsolvable.

**Strengths:**

- The paper clearly identifies a real hole in current LRM evaluations: most logic-puzzle benchmarks stay in the “canonical” zone, so they don’t reveal whether models can handle odd, rare, or slightly rule-tweaked versions.

- The benchmark spans 10 games from 6 categories (logic, grid, search, pattern, graph, sequential), so the paper’s claims aren’t tied to a single puzzle formalism. That’s good evidence that the problem is about LRMs’ robustness, not about one idiosyncratic game. It  totals >5,000 puzzles.

-  The experiments directly compare Original vs. HardcoreLogic and show that all models, including SOTA, lose accuracy when puzzles are made longer-tail.

**Weaknesses:**

- Section 2 claims that puzzles are extended along three “orthogonal” dimensions — Increased Complexity (IC: IC1, IC2), Uncommon Elements (UE: UE1, UE2), and Unsolvable Puzzles (UP). **Why are they orthogonal?**
However, Section 4 immediately notes that “puzzles may also have two different long-tail transformation attributes at the same time.”
This means the core experimental factor is conflated: when accuracy drops, we cannot uniquely attribute the drop to IC1 vs. UE1, because several knobs were turned at once. Nevertheless, Section 4.1 fits a linear model and makes a fairly strong statement that IC1 “has the greatest comprehensive impact.” That claim seems stronger than the actual design supports.
On top of that, the paper doesn’t report confidence intervals or any regularization that would stabilize the estimates, so the single set of coefficients they show looks more definite than it really is.


- In Sec. 3, “a generation run is considered correct if and only if the model successfully finishes reasoning and produces a correct answer” under a 32,768-token budget. This policy conflates two different failures: the model logically failed, and the model found the answer but over-generated verification and hit the budget.
Given that many LRMs today use self-critique / multi-step CoT, the current metric likely underestimates the real reasoning capability on the hardest instances. A secondary metric (“first correct step,” “correct intermediate state”) would make Section 3 more credible.

- The error analysis in Sec. 4.2 is a nice idea (six failure types), but it is built on a limited set of erroneous responses and on automatic judgments. With such a small pool and no human inter-annotator agreement, fine-grained statements like “stronger models show more brute-force errors” should be treated as hypotheses, not as solid findings. There is no human inter-annotator agreement reported.

- Sec. 2 says UPs “deliberately lack a valid solution” to test whether models detect inconsistency. But the main text does not spell out whether UPs are generated via contradiction, under-specification, or conflicting hybrids. In Sec. 4, however, models are evaluated on whether they “justify” unsolvability. Without a clear taxonomy of UP generation in the main paper, it is hard to know whether model mistakes are due to weak inconsistency detection or to task underspecification.




- Section 3 explicitly says that, due to cost, closed-source models are tested on a small number of cases and that per-game analysis “mainly focuses on open-source models.” Yet the intro-level narrative is still all models, including SOTA, degrade. With such small samples, long-tail variance can dominate; strong statements about closed models should be toned down or supported with stratified sampling.

**Questions:**

Please see Weaknesses

---

> ### Author Response · Authors · 2025-11-21
>
> ## A1: Transformation dimensions and cross-dimensional effects
>
> > **W1:** Why are Increased Complexity (IC: IC1, IC2), Uncommon Elements (UE: UE1, UE2), and Unsolvable Puzzles (UP) orthogonal? However, Section 4 immediately notes that “puzzles may also have two different long-tail transformation attributes at the same time.” This means the core experimental factor is conflated: when accuracy drops, we cannot uniquely attribute the drop to IC1 vs. UE1. Nevertheless, Section 4.1 fits a linear model and makes a fairly strong statement that IC1 “has the greatest comprehensive impact.” That claim seems stronger than the actual design supports. On top of that, the paper doesn't report confidence intervals or any regularization that would stabilize the estimates.
>
> 1. **["Distinct" rather than "orthogonal"]** Thank you for raising this point. Our original intention is to convey that we extend logic puzzles along **three distinct dimensions.** These dimensions are **not mutually exclusive**; each can occur independently or in combination. We have revised the wording in the manuscript to avoid misunderstanding.
>
> 2. **[We use weighted linear regression]** Regarding the analysis in Section 4.1: considering a puzzle may exhibit **multiple transformation dimensions simultaneously** (e.g., IC1 + UE1), we applied a **weighted linear regression** to estimate the contribution of each transformation type to puzzle difficulty. Concretely, we fit the following model: $$y=k_\\mathrm{IC1}\\cdot 1_\\mathrm{IC1}+k_\\mathrm{IC2}\\cdot 1_\\mathrm{IC2}+k_\\mathrm{UE1}\\cdot 1_\\mathrm{UE1}+k_\\mathrm{UE2}\\cdot 1_\\mathrm{UE2}+b$$ where $y$ is the observed accuracy for a specific puzzle variant, $1_\\mathrm{UE1}$ is a binary indicator (1 if transformation UE1 is applied, 0 otherwise), $k_\\mathrm{UE1}$ quantifies the marginal accuracy change attributable to UE1 under the assumption of additive effects, $b$ is the expected accuracy predicted by the model when all dummy variables are zero, and weights $w_i=N_i$ (sample sizes) give greater influence to observations with larger sample sizes when calculating the loss function.
>
> 3. **[Why Weighted linear regression works]** Weighted linear regression isolates the marginal effect of individual transformations through two mechanisms: (1) the additive linear model with dummy variables statistically disentangles combined transformation effects by estimating each factor's contribution relative to the baseline configuration, (2) sample-size-based weighting assigns greater influence to high-reliability observations during coefficient estimation, ensuring parameters reflect dominant patterns in robust data.
>
> 4. **[Additional analysis using single-trasnformation-only samples]** To further address the reviewer concern, we additionally utilize logic puzzles with only one transformation type to fit a model for analyzing each type's effect precisely, with more detailed analysis provided in Appendix D.5. As shown in Figure 15 in Appendix D.5 (also see **Table R1** below) , it shows quite similar results with the original analysis using all logic puzzles, showing that **IC1 has the greatest comprehensive impact.**
>
> | Game     |      IC1 |     IC2 |      UE1 | UE2 | Game        |      IC1 |      IC2 |      UE1 |      UE2 |
> | :------- | -------: | ------: | -------: | --: | :---------- | -------: | -------: | -------: | -------: |
> | Binario  | $-74.09$ |       - |        - |   - | Minesweeper | $-25.38$ |        - |        - | $-29.67$ |
> | Crypto   | $-26.58$ |       - |        - |   - | Navigation  |        - | $-10.00$ |        - |        - |
> | Hanoi    |        - |       - | $-22.51$ |   - | Skyscraper  |        - |  $-2.04$ |        - |  $-3.70$ |
> | Hitori   | $-46.06$ |       - |  $-7.00$ |   - | Sudoku      | $-53.40$ |        - | $-51.79$ | $-37.85$ |
> | Kakurasu |        - | $-9.87$ |  $-9.64$ |   - | Zebralogic  |        - | $-33.58$ |        - |        - |
>
> > **Table R1:** Single factor effects of long-tail transformations on puzzle accuracy. A bar (-) means no corresponding data for fitting.

---

> > ### Author Response · Authors · 2025-11-21
> >
> > 5. **[Statistical significance]** Additionally, we calculate the $p$-values (also see **Table R2** below) and the $95\\%$ confidence intervals of the fitted coefficients in Appendix D.5. Under the $95\\%$ confidence intervals, IC1 exhibits the largest decrease among all factors, and its $p$-values are consistently below $0.001$, indicating a highly significant effect. Taken together, these results confirm that the dominance of IC1 persists even after accounting for statistical uncertainty.
> >
> > | Puzzle   |    IC1 |    IC2 |    UE1 | UE2 | Puzzle      |    IC1 |    IC2 |    UE1 |    UE2 |
> > | :------- | :----: | :----: | :----: | :-: | :---------- | :----: | :----: | :----: | :----: |
> > | Binario  | $.000$ | $.661$ |      - |   - | Minesweeper | $.000$ |      - | $.982$ | $.000$ |
> > | Crypto   | $.000$ |      - |      - |   - | Navigation  |      - | $.132$ |      - | $.402$ |
> > | Hanoi    |      - |      - | $.000$ |   - | Skyscraper  |      - | $.098$ |      - | $.010$ |
> > | Hitori   | $.000$ |      - | $.283$ |   - | Sudoku      | $.000$ |      - | $.000$ | $.006$ |
> > | Kakurasu | $.000$ | $.188$ | $.403$ |   - | ZebraLogic  |      - | $.000$ |      - |      - |
> >
> > > **Table R2:** $p$-values of the fitted weighted linear regression on *all* samples (for Figure 6 in Section 4.1, *NOT* for Table R1). A bar (-) means no corresponding data for fitting.
> >
> > ## A2: Reasoning length constraint
> >
> > > **W2:** The $32,768$-token budget policy conflates two different failures: the model logically failed, and the model found the answer but over-generated verification and hit the budget. Given that many LRMs today use self-critique / multi-step CoT, the current metric likely underestimates the real reasoning capability on the hardest instances. A secondary metric (“first correct step,” “correct intermediate state”) would make Section 3 more credible.
> >
> > 1. **[Why choose $32,768$]** Our use of a unified $32,768$-token budget is intended to ensure **fair comparison across models with different context capacities**, similar to imposing a fixed time constraints when examining human solvers. Considering computational cost and difficulty to customize an optimal reasoning-length constraint for each specific puzzle, we therefore follow most models officical evaluation protocal and also prior logic-puzzle benchmarks [1,2] in adopting a fixed $32,768$-token limit.
> >
> >    [1] Lin et. al., *ZebraLogic: On the Scaling Limits of LLMs for Logical Reasoning.*
> >
> >    [2] Chen et. al., *Enigmata: Scaling Logical Reasoning in Large Language Models with Synthetic Verifiable Puzzles.*
> >
> > 2. **[Length-exceeding Statistics]** We report the ratio of length-exceeding cases across all models for each game in **Table R3**. Only Skyscraper, Sudoku and Binario have more than $15\\%$ (and still less than $30\\%$) cases that failed to finish reasoning in $32,768$ tokens, while their average accuracy are $2.32\\%$, $16.29\\%$ and $20.96\\%$. This indicates that the vast majority of failures, even on hard games, are not caused by hitting the token budget.
> >
> > | Game     |  LE (%) | Game        |  LE (%) |
> > | :------- | ------: | :---------- | ------: |
> > | Binario  | $22.31$ | Minesweeper | $11.08$ |
> > | Crypto   |  $5.85$ | Navigation  |  $4.94$ |
> > | Hanoi    |  $4.56$ | Skyscraper  | $28.26$ |
> > | Hitori   |  $7.86$ | Sudoku      | $22.65$ |
> > | Kakurasu |  $2.48$ | ZebraLogic  |  $9.68$ |
> >
> > > **Table R3:** Ratio of length-exceeding (LE) cases across all models for each game.

---

> > > ### Author Response · Authors · 2025-11-21
> > >
> > > 3. **[Length-Exceeding Category]** Furthermore, to better understand model failures in length-exceeding cases for four representative models (as shown in **Table R4**), we categorize such cases into four interpretable and mutually exclusive types:
> > >
> > >    - Brute Force: the model attempts to enumerate possibilities or perform unstructured brute-force exploration, resulting in excessive, inefficient reasoning steps that exceed the allowed length.
> > >    - Over Verification: the model has already produced a correct solution at some point but subsequently engages in self-doubt or unnecessary self-critique, ultimately continuing to reason far beyond the point of correctness and exceeding the output limit.
> > >    - Infinite Repetition: the model enters repetitive, meaningless loops, producing output that grows without contributing any additional logical progress.
> > >    - Incomplete Reasoning: the model is still in the middle of its reasoning chain when the output terminates.(the model continues exploring different strategies or deductions without resolving the puzzle)
> > >
> > >    We can observe that Over Verification and Incomplete Reasoning account for only a small fraction of length-exceeding cases across all models. In contrast, the majority of such failures are attributable to either Brute Force behavior where the model attempts an unbounded search or Infinite Repetition, where the reasoning process collapses into non-meaningful loops.
> > >
> > > | Error Type           | gpt-oss-120b | Qwen3-235B-A22B | Kimi-K2 | Minimax-M1 |
> > > | :------------------- | -----------: | --------------: | ------: | ---------: |
> > > | Brute Force          |       $84.0$ |          $78.0$ |   $0.0$ |     $18.0$ |
> > > | Over Verfication     |        $4.0$ |           $0.0$ |   $0.0$ |      $0.0$ |
> > > | Infinite Repetition  |        $0.0$ |           $6.0$ | $100.0$ |     $78.0$ |
> > > | Incomplete Reasoning |       $12.0$ |          $16.0$ |   $0.0$ |      $4.0$ |
> > >
> > > > **Table R4:** Error type distribution (%) on length-exceeding cases of several models.
> > >
> > > 4. **[Extending Token Budget Experiments]** To further address the reviewer's concern, we extend the reasoning length constraint to $65,536$ tokens for **gpt-oss-120b** and **seed-oss-120b** and test them on the three hard games mentioned above (Skyscraper, Sudoku and Binario) that have most length-exceeding cases. Results are reported in **Table R5**. While increasing the token budget does yield modest performance gain when the accuracy is low, it also doubles the inference cost. More importantly, increasing the token budget also boosts Original accuracy, meaning that the relative gap between Original and HardcoreLogic remains nearly unchanged. These findings confirm that the observed performance degradation is not primarily caused by length constraints, but arises from the increased difficulty introduced by our puzzle transformations.
> > >
> > > | Game       | Benchmark     | gpt-oss-120b | (65,536) | seed-oss-36b | (65,536) |
> > > | :--------- | :------------ | -----------: | -------: | -----------: | -------: |
> > > | Sudoku     | Original      |      $97.25$ |  $97.00$ |      $73.50$ |  $78.75$ |
> > > |            | HardcoreLogic |      $58.22$ |  $59.83$ |      $24.17$ |  $28.94$ |
> > > |            | **Gap**       |     $-39.03$ | $-37.17$ |     $-49.33$ | $-49.81$ |
> > > | Skyscraper | Original      |      $15.50$ |  $17.12$ |      $10.62$ |  $17.25$ |
> > > |            | HardcoreLogic |       $9.04$ |   $9.38$ |       $4.71$ |   $7.00$ |
> > > |            | **Gap**       |      $-6.46$ |  $-7.74$ |      $-5.91$ | $-10.25$ |
> > > | Binario    | Original      |      $99.00$ |  $99.33$ |      $99.66$ |  $99.50$ |
> > > |            | HardcoreLogic |      $42.67$ |  $44.58$ |      $31.67$ |  $32.92$ |
> > > |            | **Gap**       |     $-56.33$ | $-54.75$ |     $-67.99$ | $-66.58$ |
> > >
> > > > **Table R5:** Model accuracy (%) of **gpt-oss-120b** and **seed-oss-36b** on Sudoku, Skyscraper, and Binario under two reasoning-length budgets ($32,768$ on model-name columns, $65,536$ tokens on columns to their right). For each game, we report results on both Original and HardcoreLogic, as well as the gap between them.
> > >
> > > 5. **[Intermedaite Measurements are not reliable metrics]** Internal reasoning steps or intermediate results that appear to lead toward a correct answer do not necessarily reflect the model's final output, as LLMs tends to revise their proposed answers or they are not fully faithful. As highlighted in the error analysis in Section 4.2, strong models such as **gpt-oss-120b** and **qwen3-235b** still exhibit a notable portion of **Over Verification** errors (approximately $16\\%$ and $12\\%$ of their reasoning failures), where the model reaches the correct answer during reasoning but subsequently overturns it due to self-doubt or unnecessary self-critique. Therefore, we rely on the current end-to-end rule to judge the models' performance.

---

> > > > ### Author Response · Authors · 2025-11-21
> > > >
> > > > ## A3: Error analysis
> > > >
> > > > > **W3:** The error analysis in Sec. 4.2 is a nice idea (six failure types), but it is built on a limited set of erroneous responses and on automatic judgments. With such a small pool and no human inter-annotator agreement, fine-grained statements like “stronger models show more brute-force errors” should be treated as hypotheses, not as solid findings. There is no human inter-annotator agreement reported.
> > > >
> > > > HardcoreLogic contains over $5,000$ extremely challenging puzzles, and model outputs are often extremely long, making full manual inspection infeasible.  Even with assistance from LLMs, analyzing every erroneous response across all models and games would far exceed our annotation budget. Therefore, the error analysis in Section 4.2 is conducted on a uniformly sampled subset rather than the full dataset ($50$ erroneous cases from each model for analyzing both Original and Hardcore datasets, $5$ cases per game).
> > > >
> > > > To improve the reliability of error annotations, we extend beyond GPT-5 and incorporate **two additional LLM annotators**, including **Gemini-2.5 Pro** and **Claude Sonnet 4.5**. For each sampled error, we accept the label when at least two models agree; otherwise, we perform manual adjudication. Across the three LLM annotators, we obtain a **Fleiss'** $\\kappa$ of $0.47$, indicating moderate agreement.
> > > >
> > > > **Table R6** reports the updated error analysis using this multi-annotator scheme for both Original and HardcoreLogic datasets. Importantly, the new error distributions (three LLMs-human hybrid) closely match those produced by GPT-5 alone: the two annotation sets yield a **Cohen's** $\\kappa$ of $0.81$, reflecting high consistency. According to the updated annotations, we observe that stronger models such as **gpt-oss-120b** and **Qwen3-235B-A22B** still exhibit around $28\\%$ of their failures due to brute-force reasoning, which aligns closely with the proportions identified by GPT-5 previously. We have incorporated new annotation schema and updated results in Appendix D.3  of the revised manuscript.
> > > >
> > > > | Benchmark     | Error Type          | gpt-oss-120b | Qwen3-235B-A22B | Kimi-K2 | Minimax-M1 |
> > > > | :------------ | :------------------ | -----------: | --------------: | ------: | ---------: |
> > > > | Original      | Misunderstanding    |        $2.0$ |           $2.0$ |   $8.0$ |      $2.0$ |
> > > > |               | Misapplied          |        $2.0$ |           $2.0$ |   $4.0$ |      $0.0$ |
> > > > |               | Brute Force         |       $20.0$ |          $34.0$ |  $10.0$ |     $22.0$ |
> > > > |               | Factual Errors      |       $50.0$ |          $38.0$ |  $56.0$ |     $20.0$ |
> > > > |               | Over Verification   |       $24.0$ |          $24.0$ |  $20.0$ |      $0.0$ |
> > > > |               | Infinite Repetition |        $2.0$ |           $0.0$ |   $0.0$ |     $56.0$ |
> > > > | HardcoreLogic | Misunderstanding    |       $10.0$ |          $12.0$ |  $22.0$ |      $2.0$ |
> > > > |               | Misapplied          |        $8.0$ |          $12.0$ |  $14.0$ |      $2.0$ |
> > > > |               | Brute Force         |       $28.0$ |          $28.0$ |   $4.0$ |     $32.0$ |
> > > > |               | Factual Errors      |       $38.0$ |          $34.0$ |  $58.0$ |     $24.0$ |
> > > > |               | Over Verification   |       $16.0$ |          $12.0$ |   $2.0$ |      $2.0$ |
> > > > |               | Infinite Repetition |        $0.0$ |           $2.0$ |   $0.0$ |     $38.0$ |
> > > >
> > > > > **Table R6:** Error type distribution (%) on $50$ error samples per model, integrating annotations from three annotating LLMs and human adjudication.

---

> > > > > ### Author Response · Authors · 2025-11-21
> > > > >
> > > > > ## A4: Unsolvable puzzle
> > > > >
> > > > > > **W4:** The main text does not spell out whether UPs are generated via contradiction, under-specification, or conflicting hybrids. In Sec. 4, however, models  are evaluated on whether they “justify” unsolvability. Without a clear taxonomy of UP generation in the main paper, it is hard to know whether model mistakes are due to weak inconsistency detection or to task underspecification.
> > > > >
> > > > > We apologize for causing confusion regarding the unsolvable puzzle construction. We have provided detailed UP transformations for each game in Appendices B.2 and B.3. In summary, among the three mechanisms the reviewer mentioned, our UP transformations only rely on injecting contradictions among constraints. We do not adopt under-specification because it usually leads to multiple plausable solutions due to ambiguity rather than having no valid solution, therefore our objective is to test whether models detect inconsistency.
> > > > >
> > > > > ## A5: Results on closed-source model
> > > > >
> > > > > > **W5:** Section 3 explicitly says that, due to cost, closed-source models are tested on a small number of cases and that per-game analysis “mainly focuses on open-source models.” Yet the intro-level narrative is still all models, including SOTA, degrade. With such small samples, long-tail variance can dominate; strong statements about closed models should be toned down or supported with stratified sampling.
> > > > >
> > > > > We would like to clarify that the sample size for closed-source evaluation is $600$ **instances, which is not small.** Specifically, we sample $5$ cases for each subtasks and we want to emphasize that **each game contains multiple subtasks** corresponding to different transformations. Across the $10$ games, there are $120$ subtasks in total, yielding $5\\times 120=600$ sampled instances for evaluating closed-source models instead of $5\\times 10=50$.  In context that many widely used reasoning benchmarks contain only a few hundred samples, such as Math500 and GPQA, we believe that $600$ samples across diverse long-tail transformations are sufficient to support overall assessment of closed-source models. (We agree that this sample size is not adequate for **per-game** conclusions, which is why closed-source models are excluded from per-game analyses). We have clarified these details in Section 3.1 to avoid misunderstanding.

---

### Author Response · Authors · 2025-11-21
**General response to reviewers and AC (and thank you for your feedback and supervision!)**

We are grateful to all reviewers for their valuable feedback and recognition of the significance of our work, including **addressing a critical gap in LRM evaluation** (`na2W`, `izsk`), **providing a valuable and challenging benchmark** (`izsk`, `xWwM`, `eXAq`), **systematic construction and scope coverage** (`na2W`, `eXAq`), **inclusion of unsolvable instances** (`izsk`), **thorough and insightful analysis** (`na2W`, `izsk`, `eXAq`) and **high presentation quality** (`izsk`, `eXAq`).

In response to the reviewers' comments and questions, we provide rebuttal arguments including (but not limited to):

- Clarification of **transformation dimensions** and **cross-dimensional effects** (`na2W`);
- Additional analysis regarding **reasoning length constraint** (`na2W`);
- **Strengthened error analysis** with more details (`na2W`, `izsk`, `xWwM`);
- Further explanation on unsolvable puzzles and results on closed-source model (`na2W`);
- Introduction to **complexity metrics** and their **relation to empirical difficulty** (`izsk`);
- Explanation of **data construction** and **output evaluation** procedures (`xWwM`);
- **Detailed emphasis on the novelty of our work** (`xWwM`);
- Quantitative analysis of **transformation comprehensibility** (`eXAq`);
- Clarification of the experiment results on Skyscraper (`eXAq`).

We hope these responses address your key concerns, and we welcome any further feedback or clarification requests, for which we are glad to provide necessary additional information.

---

### Meta-Review · Area_Chair_sYUT · 2026-01-10

**Summary:**

The primary concerns raised by reviewers focused on the reliability of the automated error analysis, the independence of the defined transformation dimensions, and specific experimental design choices such as the fixed token budget. Reviewers specifically questioned whether the error taxonomy relied too heavily on GPT-5 without human verification and whether the dimensions of difficulty (Increased Complexity, Uncommon Elements, Unsolvable Puzzles) were conflated. Additionally, clarity regarding the data construction pipeline, particularly for unsolvable puzzles, and the correlation between theoretical complexity metrics and empirical difficulty was requested. The authors effectively addressed these points during the rebuttal by incorporating a multi-model annotation scheme with human adjudication to validate error analysis, providing weighted regression analyses to disentangle transformation effects, and presenting supplementary data to rule out token limitations as a primary failure mode. The successful addressing of these concerns, combined with the benchmark's utility in evaluating robust reasoning beyond memorization, supports the recommendation for acceptance.

**Reviewer Concerns:**

The rebuttal addressed the technical and methodological concerns raised by the reviewers. The skepticism regarding the reliability of the automated error analysis was addressed by the authors' new multi-model annotation pipeline with human adjudication, which confirmed the validity of the initial findings. The concerns about the statistical independence of transformation dimensions were addressed through the introduction of weighted linear regression analyses and confidence intervals, while the hypothesis that token budget constraints were a confounding factor was ruled out by supplementary experiments showing minimal performance gains with doubled token limits. Moreover, questions regarding the clarity of data construction and potential ambiguity in puzzle transformations were answered with detailed explanations of the rule-based validation processes. While the concern regarding the fundamental novelty of the benchmark represents a subjective difference of opinion rather than a technical flaw, the authors provided a strong justification for the necessity of long-tail evaluations to distinguish genuine reasoning from memorization.

**Reviewer Scores:**

Reviewer eXAq engaged in the discussion and maintained their high score. For the remaining reviewers who did not respond to the authors' rebuttal, it is highly probable that Reviewers na2W and xWwM would have raised their scores from 4 (marginally below acceptance) to 6 (weak accept) or higher. Reviewer na2W's primary concerns regarding the independence of transformation dimensions and the token budget were directly addressed by the new weighted regression analysis and the supplementary experiments showing minimal impact of increased token limits. Reviewer xWwM's doubts about data construction clarity and error analysis reliability were addressed by the detailed breakdown of the generation pipeline and the introduction of the multi-model annotation scheme. Reviewer izsk, who already assigned a score of 8, would likely have maintained this strong rating while increasing their confidence, as their specific requests for correlation analysis between theoretical complexity and empirical difficulty were met with the new statistical data provided in the rebuttal.

---

### Decision · Program_Chairs · 2026-01-26

Accept (Poster)